# High-order tensor flow processing using integrated photonic circuits

Shaofu Xu[1,2], Jing Wang[1,2], Sicheng Yi[1] & Weiwen Zou [1] ✉

Tensor analytics lays the mathematical basis for the prosperous promotion of multiway signal processing. To increase computing throughput, mainstream processors transform tensor convolutions into matrix multiplications to enhance the parallelism of computing. However, such order-reducing transformation produces data duplicates and consumes additional memory. Here, we propose an integrated photonic tensor flow processor (PTFP) without digitally duplicating the input data. It outputs the convolved tensor as the input tensor 'flows' through the processor. The hybrid manipulation of optical wavelengths, space dimensions, and time delay steps, enables the direct representation and processing of high-order tensors in the optical domain. In the proof-of-concept experiment, an integrated processor manipulating wavelengths and delay steps is implemented for demonstrating the key functionalities of PTFP. The multi-channel images and videos are processed at the modulation rate of 20 Gbaud. A convolutional neural network for video action recognition is demonstrated on the processor, which achieves an accuracy of 97.9%.

Stacking data of multiple dimensions to form a tensor provides us the opportunity to discover the intrinsic structural features hidden in the data[1], which are invisible from two-way (matrix) data analysis. For example, multiway representation of electroencephalogram (EEG) data is the natural and effective way of neuroscience data processing[2]. The tensor stacked across time, space, and spectrum is beneficial to detect features in electromagnetic waveforms[3]. Since tensor matches the high-dimensional nature of the world, the concept of multiway analytics gives rise to extensive signal processing approaches in fields including life science[2,4], radar[5,6], data mining[7,8], and machine learning[9–11]. Among the basic operations for tensors, convolution effectively extracts structural features from data. Targeted features are filtered out as the convolutional kernel traverses the tensor. As an epitome, a convolutional neural network, which plays a fundamental role in modern artificial intelligence (AI), is designed under the concept of multi-channel tensor processing[12,13].

Given the fact that tensor convolution, especially in the AI field, is consuming an increasing portion of computing resources, high-throughput and energy-efficient processors are desired[14]. Digital methods including generalized matrix multiplication (GeMM)[15], domain transformation[16], and input/weight reusing[17] are investigated to achieve high-performance computing (HPC) of tensor convolution. These methods pursue a balanced and optimized performance under limited hardware resources (e.g. memory, bandwidth, and power). Among these methods, GeMM is widely adopted for its high throughput and high flexibility for AI. For example, in the Tensor Core of Nvidia Ampere architecture[18], the CUBE core of Huawei Davinci architecture[19], the systolic array of Google TPU architecture[20], and the cross-bar array of memristor architecture[21], high-order tensor convolutions are transformed to two-dimensional matrix multiplications so that paralleled computational cores can work simultaneously to enhance throughput. However, during the GeMM transformation, the input tensor should be duplicated and shifted multiple times (related to the kernel size) to form an input matrix, which significantly increases memory use and additional memory access.

[1]State Key Laboratory of Advanced Optical Communication Systems and Networks, Intelligent Microwave Lightwave Integration Innovation Center (imLic), Department of Electronic Engineering, Shanghai Jiao Tong University, 200240 Shanghai, China. [2]These authors contributed equally: Shaofu Xu, Jing Wang. ✉e-mail: wzou@sjtu.edu.cn

Besides electronic HPC processors, photonics is recently demonstrated as a promising candidate to build high-performance matrix processors. By designing the photonic circuit as linear transformation functions, matrix multiplications can be accomplished as the light flies through the circuit[22–24]. The broadband spectrum of photonic circuits boosts the clock frequency to tens of Gigahertz $(10^9\,\mathrm{Hz})$[25–27]. Consequently, photonic circuits are demonstrated as superior GeMM processors with high throughput and energy efficiency[28,29]. Another advantage of photonics compared with electronics is that the available degrees of freedom (DoF) of light are rich. For example, wavelengths[23,26,29], guiding modes[30], time[31], and space[22,24,32] are successfully investigated to carry out linear transformations. Although in theory[33,34], we can take a hybrid use of multiple DoF of light to expand the representation dimensionality of photonics so that high-order tensors can be directly processed instead of using the GeMM, such a photonic processer has not been demonstrated.

Here, we propose an integrated photonic tensor flow processer (PTFP) that processes high-order tensors without digital data duplication and shifting; therefore, excess memory is saved for input data preparation. The serially input data directly enter the PTFP and the output result is yielded serially. Namely, tensor convolution is completed as the input tensor 'flows' through the photonic circuit. This is achieved by the hybrid manipulation of optical wavelengths, time delays, and space dimensions. Kernel weights are implemented inside the microring resonators (MRRs) of the PTFP and data registering is accomplished by the embedded optical delay structure. In a proof-of-concept experiment, we implement a silicon-based integrated photonic chip to conduct the key functionalities of the PTFP, i.e. the hybrid manipulation of wavelengths and time delays. It demonstrates two-order tensor flow processing and reduces memory use 3 times. Improving the integration scale will upgrade it as a four-order tensor processor and promote memory use reduction. Empowered by the broadband capability of light, the photonic chip works at the modulation rate of 20 Gbaud and is capable to achieve a compute density surpassing trillions of operations per second per square millimeter. By reconfiguring the weights of MRRs and reusing the PTFP chip, tensor (including multi-channel images and video) processing is experimentally demonstrated. A CNN for action recognition is trained to validate the PTFP chip. An accuracy of 97.9 % on the KTH dataset[35] is achieved at the inference phase.

Basic principles of the GeMM and the PTFP are compared in Fig. 1a. The dimensionality of the input tensor is denoted as $[C_\mathrm{in}, D_\mathrm{data}]$, where $C_\mathrm{in}$ denotes the number of input channels and $D_\mathrm{data}$ is the size of data in a single input channel (e.g. $D_\mathrm{data}$ is $[L, L]$ when the input is an image with a lateral size of $L$). Different from the conventional convolution, tensor convolution with multiple input channels should yield multiple output channels. $[C_\mathrm{out}, D_\mathrm{data}]$ denotes the dimensionality of the output tensor. Each output channel is obtained by summing all convolved results from every input channel. Therefore, the dimensionality of a complete kernel of tensor convolution is denoted as $[D_\mathrm{kernel}, C_\mathrm{in}, C_\mathrm{out}]$, where $D_\mathrm{kernel}$ is the size of a single convolution. In order to compute tensor convolution with the stride of '1', shown by the 'GeMM' part of Fig. 1a, GeMM firstly transforms the input tensor to an input matrix with the dimensionality of $[C_\mathrm{in} \times D_\mathrm{kernel}, D_\mathrm{data}]$, where data volume is augmented by $D_\mathrm{kernel}$ times. The additional data is generated by duplicating and shifting the original data, occupying more memories and taking more memory access. The kernel tensor is reshaped to a two-dimensional matrix $[C_\mathrm{out}, C_\mathrm{in} \times D_\mathrm{kernel}]$. Then, the output tensor is obtained by matrix multiplication. In the process of the PTFP (shown in the 'Flow' part), the input tensor is not duplicated. Different input channels are carried by different optical wavelengths. Serial pixels in a single channel are temporally modulated onto the time steps of an optical signal. Inside the PTFP, each input channel is connected with each output channel through a convolutional operation (a line in the figure). A convolutional operation is essentially a

finite impulse response (FIR) filter; therefore, we can implement such FIR filters by imposing delaying, weighting, and summation to the input temporal sequence. The number of delay steps is equal to the size of kernel, $D_\mathrm{kernel}$. The additional memory required by GeMM is equivalently accomplished with the optical delay structure. In other words, the PTFP approach saves $D_\mathrm{kernel}$ times of digital memory for input tensor transformation. The size of memory for weights is the same as conventional GeMM. Given that the input sequences are carried on different wavelengths, the convolved sequences are combined to yield an output channel with wavelength division multiplexing (WDM). Other output channels (depicted with gray lines) are similarly yielded by spatially duplicating the same structure but configuring different kernel weights.

Following the PTFP concept, the schematic of an integrated chip is illustrated in Fig. 1b. Input optical sequences of different wavelengths are first combined with a WDM. Then, directional couplers and ODLs are deployed to provide the time delay steps $D_\mathrm{t}$ $(=D_\mathrm{kernel})$. In each delay step, optical sequences are further split to provide the dimension of space, $D_\mathrm{s}$. In a specific delay step and space dimension, a weighting bank with $D_\mathrm{w}$ $(=C_\mathrm{in})$ copies MRRs is exploited. A single MRR in a weighting bank controls the transmission rate of a specific wavelength. By shifting the resonance wavelength of MRRs, weights of input wavelengths can be reconfigured. Via these $D_\mathrm{w} \times D_\mathrm{t} \times D_\mathrm{s}$ copies of MRRs, multiplications involved in a complete convolutional kernel are accomplished. After weighting, photodetectors (PDs) convert the total optical power of all wavelengths to electrical signals, performing summation across different input channels. And the electrical power combiners (EPCs) perform electrical summations of signals across different delay steps. Since operations on the chip are linear, two steps of summations are commutative. Every output sequence of the EPC corresponds to an output channel in Fig. 1a.

## Results
### The fabricated PTFP chip
Figure 2a shows the photograph of the PTFP chip, which is fabricated with a standard Silicon-on-Insulator (SOI) integration process. As proof-of-concept, we implement the key components of the PTFP onto the chip, including WDM, optical delays, and weighting banks. The fabricated chip conducts four-channel convolution with three parameters in each channel, namely a two-order tensor convolution kernel written as $[D_\mathrm{kernel} = \mathrm{height} \times \mathrm{width} = 1 \times 3, C_\mathrm{in} = 4, C_\mathrm{out} = 1]$. Given the fact that duplicating the same structure can expand space dimension $(C_\mathrm{out})$ and cascading more ODLs will expand 'height' and 'width' dimensions[33,34], successful validation of this chip constructs a strong basis for a complete four-order tensor convolution processor. Optical signals enter and leave the chip through the waveguide-fiber edge coupler array. Figure 2b depicts the layout of the fabricated PTFP chip, comprising a four-way WDM, two cascaded ODLs, and three weighting banks with four MRRs inside each. The WDM shown in Fig. 2c is designed with the asymmetric Mach–Zehnder interferometer structure. Figure 2d presents the transmission rate measurement of the WDM, showing 2-nm channel spacing and <1.2 dB channel flatness within a free spectral range (FSR). In the experiment, we choose four wavelengths located at 1550.8, 1552.8, 1554.8, and 1556.8 nm to ensure that all operating wavelengths are within the flat band of the WDM. Figure 2e, f illustrates the photograph and characterization result of a weighting bank. By increasing the voltage on the MRR, the resonating wavelength is red-shifted. Since the operating wavelengths are fixed, the variation of the MRR transmission rate performs as a weighting factor to the specific wavelength. Figure 2g provides the normalized weights (transmission rate) of every MRR with the variation of applied voltages. This weight–voltage mapping is measured once under static temperature and is used for translating kernel weights to the applied voltages. Detailed information on photonic device design and characterization can be found in Supplementary notes 1–5.

## Tensor convolution

To validate the tensor processing capability of the PTFP chip, we carry out an experiment with multi-channel images as the input tensor. Figure 3a illustrates the conceptual experimental setup (see the "Methods" section for details of the experimental setup). The PTFP chip accepts four input signals with different wavelengths. Each signal represents an input channel, i.e. an image of the multi-channel images. These images are loaded onto optical intensities via temporal modulation row by row. Four-way signals are generated with the symbol rate of 20 Gbaud, also known as the clock frequency of 20 GHz. Since the optical intensities of different wavelengths are summed up in the PD, it is necessary to carry out input synchronization to avoid symbol misalignment. Similarly, the output signals with different optical delays should be also synchronized since they are summed up in the EPC. We deploy tunable delay lines before and after the optical ports of the PTFP chip for synchronization. Figure 3b shows the result of output synchronization. In this measurement, only one input channel is adopted, so the output waveform should be identical except for delay. We observe that, after synchronization, the delay difference of every output waveform is 50 ps, corresponding to the symbol rate of

20 Gbaud. Using one input channel, we can conduct $1 \times 3$ convolutions by applying weights to the MRRs. Figure 3c is an example of the convolved waveform. The applied weights are [−1, 0, 1]. From the zoom-in plot, we observe that the experimental results are close to the theoretically calculated samples, verifying the correctness of conducting one-dimensional convolution. The deviation between the experimental result and the calculated one is mainly caused by experimental noise and waveform distortion.

As we have multiple channels for input, we validate the multi-channel convolution in this part. Three different images from a 'traffic camera' dataset[36] are chosen as the input channels and a vertical edge detection kernel [−1, 0, 1] is adopted for each of them. Therefore, the output of the multi-channel convolution should be the superposition of the vertical edges of these images. Because 3 delay steps are implemented on the chip, these $1 \times 3$ convolutions are performed without input data duplication. Figure 3d depicts the result. Three images including a car in each are processed by three wavelength channels and the output shows all vertical edges of these cars. The 'leaves' on the ground are static for three images. The vertical edges of them accumulate three times so that they are very bright in the output.

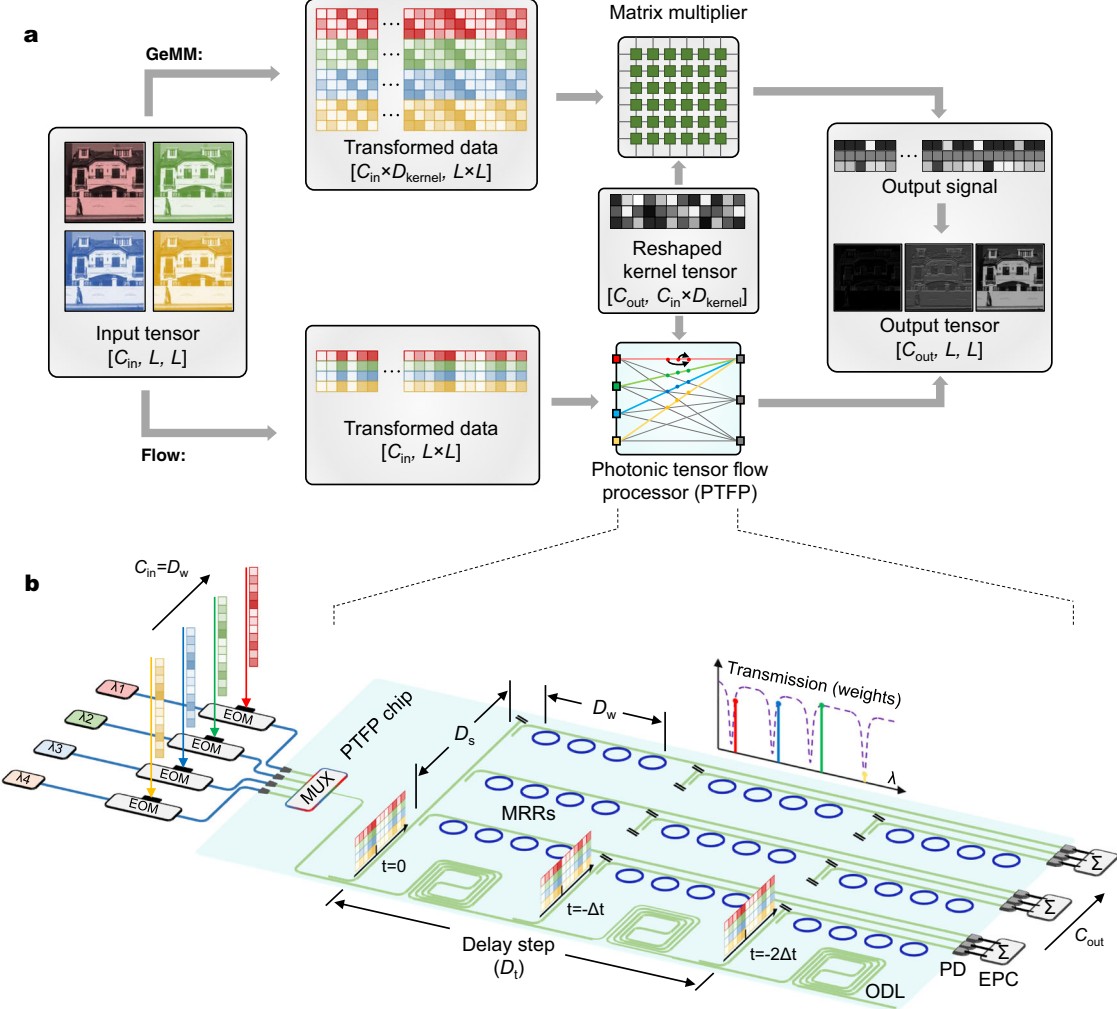

**Fig. 1 | Basic principles of the PTFP. a** Principles of the conventional GeMM and the PTFP. Before matrix multiplication, the GeMM reshapes and duplicates the input tensor ($[C_{in}, L, L]$) to the transformed data ($[C_{in} \times D_{kernel}, L \times L]$). Input data is duplicated $D_{kernel}$ times. After the matrix multiplier, outputs are yielded and can be reconstructed to convolved feature maps. In the PTFP approach (marked with 'Flow'), the input tensor is reshaped and enters the PTFP in serial. Each input channel is temporally modulated onto an individual wavelength. A line in the PTFP schematic represents a convolutional operation between an input channel and an output channel. Inside each line, signals are delayed, weighted, and summed so that a temporal convolution (a.k.a. FIR filter) is completed. An output channel is yielded by combining convolved signals from different input channels. Other output channels can be realized by spatially duplicating the same structure. **b** Conceptual schematic of the PTFP chip. EOM electro-optic modulator, MUX wavelength multiplexer, ODL optical delay line. The directional couplers and delay lines perform data duplication and shifting in the optical domain. Multiple wavelengths are split and delayed in parallel. Crossing waveguides are virtually broken for the succinctness of the graph.

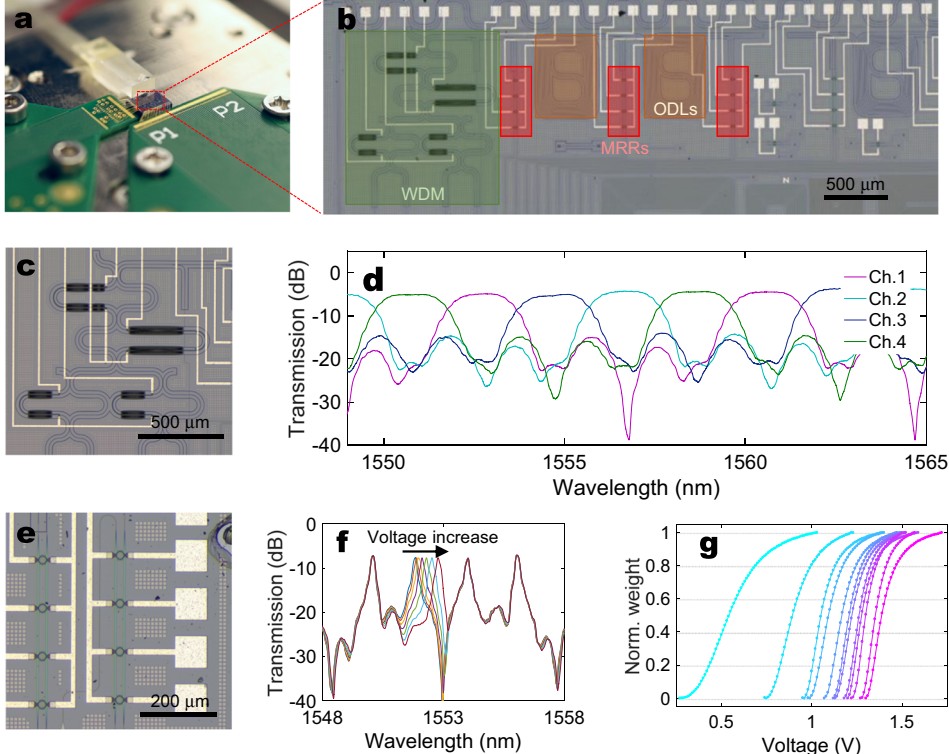

**Fig. 2 | Chip fabrication and characterization. a** Photograph of the packaged PTFP chip. Optical signals enter and leave the chip via an edge-coupled fiber array. **b** Layout of the PTFP chip. Four wavelengths are combined in the WDM. Two optical delay lines (ODLs) are deployed to provide three delay steps. Before and after each ODL, weighting banks with four MRRs in each are implemented. **c** Photograph of the WDM. **d** Transmission spectra of the WDM. **e** Regional photograph of the MRR array. **f** Transmission spectrum of the MRR array. Different voltages (0–1400 mV with 200 mV/step) are applied on the second MRR. A similar result can be obtained when voltage is applied to other MRRs. **g** Transmission rate of all 12 MRRs on the chip under voltage tuning. These curves represent weight–voltage mappings after normalization. The original resonance points of MRRs are different because of fabrication deviation.

If we configure three input channels with an identical image with row shifting, the multi-channel convolution can equivalently perform a 3 × 3 convolution on a single image. We should note that in this configuration, the input image is digitally duplicated three times since the on-chip delay structure only reduces digital memory use from 9 copies to 3 copies. The equivalent 3 × 3 convolution is a good way to benchmark the multi-channel convolution of this chip and several results are shown in Fig. 3e–i. The horizontal Sobel kernel extracts grayscale variations along the horizontal direction, so the convolved image is composed of vertical edges. Similarly, the vertical Sobel kernel can extract the horizontal edges of the image. A kernel with nine same weights can blur the image. When a Sobel kernel is superposed with an identical kernel, the image can be sharpened and the edge contrast is increased. The experimental results verify the capability of the PTFP chip to conduct multi-channel convolution.

**Human action recognition using the PTFP chip**

Based on the successful validation of multi-channel image convolution, we move forward to implement a CNN to recognize human actions in the KTH dataset. Figure 4a gives the structure of the built CNN with two convolutional layers, a recurrent layer, and a fully connected layer. We generate 4998 video segments from the KTH dataset (see the "Methods" section for video preprocessing). 3998 segments out of them are randomly picked as trainset and the left 1000 segments are used as testset. The parameters of CNN are trained firstly on a computer and the PTFP chip is used for computing the 'Conv. 1' and 'Conv. 2' layers in the inference phase (details of the training are provided in the "Methods" section). Five frames of video are input into the neural network as the input tensor. Similar to the experiment of image convolution, each input frame is reshaped to a row vector for temporal

modulation. For the first convolutional layer, the adopted kernel size is [1 × 3 × 3, 1, 4]. Given that the fabricated chip is smaller than the kernel size, the kernel is decomposed into small parts and calculated by recalling the PTFP chip multiple times. The PTFP chip calculates a kernel of [1 × 3, 3, 1] for each time of recalling and accomplishes the complete kernel for 4 times of recalling. The same decomposition method is used for calculating the second convolutional layer with a kernel size of [1 × 3 × 3, 4, 8]. Figure 4b and c display several experimental results of the first convolutional layer and the second convolutional layer, respectively. We observe that the convolved frames output by the PTFP chip is consistent with that of a digital computer, except for some experimental noise. These two convolutional layers extract frame features that contribute to action recognition. By finishing the following nonlinear layer, the recurrent layer, and the fully connected layer in an auxiliary computing device, a recognition result is obtained. The diffusion matrix with five categories of human actions ('boxing', 'handwaving', 'handclapping', 'walking', and 'running') is shown in Fig. 4d and a reference is offered in Fig. 4e. Ninety-six video segments randomly selected from the testset are recognized. Numbers on the diagonal line count correct recognition. It is shown that the recognition accuracy of the PTFP chip is 94/96 = 97.9% and that of a digital computer is 95/96 = 98.9%. The recognition result confirms that the PTFP chip accomplishes tensor convolution successfully. In Fig. 4f, we carry out simulations to reveal how the noise in the output signals affects the recognition accuracy (see the "Methods" section). Consistent with intuition, the accuracy tends to decrease with large noise amplitude. The standard deviation of the experimental error is around 0.1 and the achieved accuracy is slightly higher than the situation with pure Gaussian noise at the same level ($\sigma_{noise} = 0.1$).

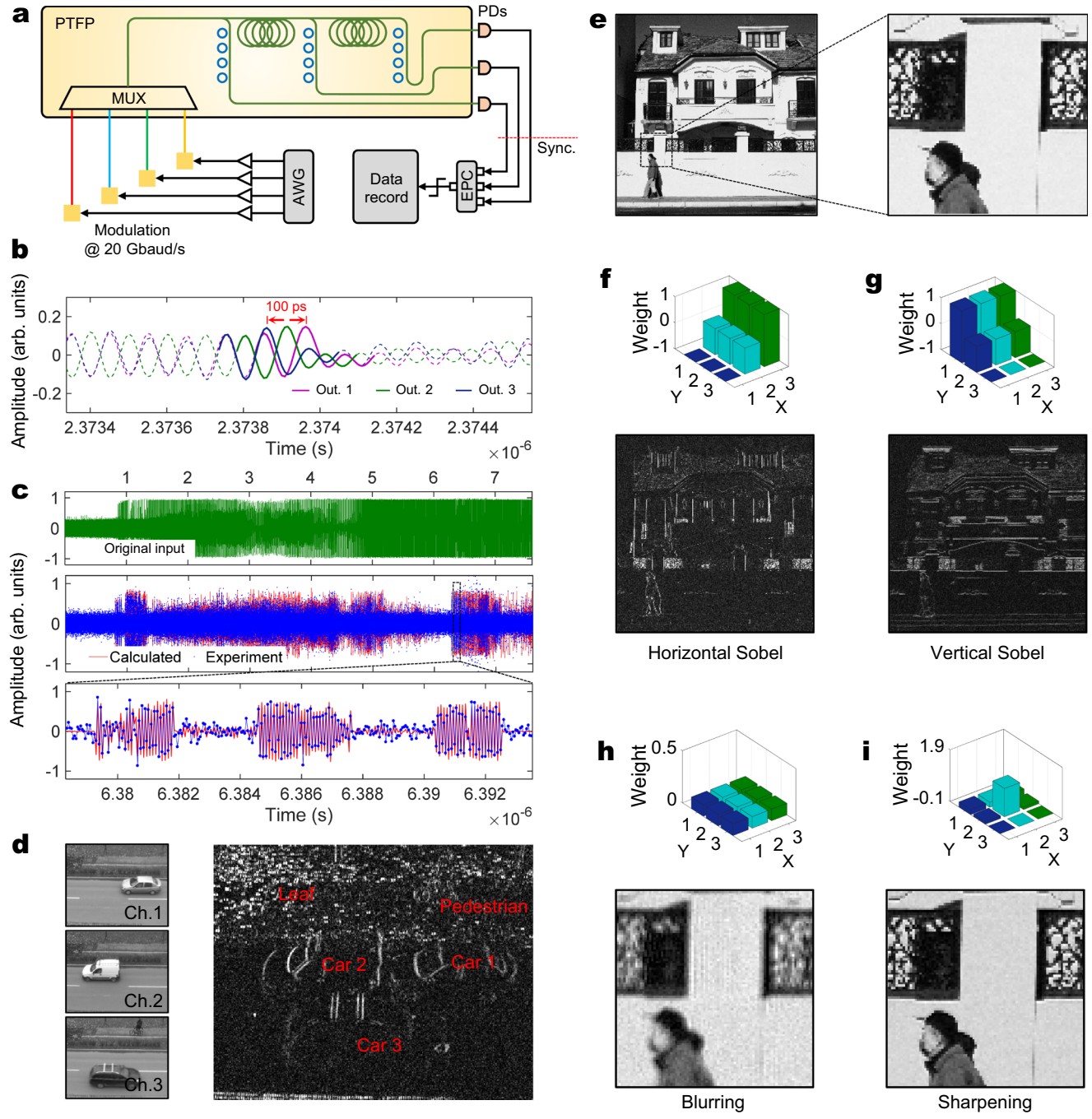

**Fig. 3 | Experimental results of tensor convolution. a** Conceptual experimental setup of the PTFP chip. MUX, wavelength multiplexer; AWG, arbitrary waveform generator. The generated signals are amplified and modulated on the optical carrier. After the tensor convolution, digital data is recorded at the output of the EPC. **b** Output synchronization. In synchronization, only one waveform is used to input one signal. The waveforms of different output ports are identical with different delays. We highlight an identical segment of these waveforms with thicker linewidth. **c** Output samples of the 1 × 3 convolutions. The original input is given for reference and a zoom-in plot is given for details. **d** Result of multi-channel convolution. The input images are listed as left insets. The result is depicted on the right. Typical objects are labeled. **e** The original image for demonstrating multi-channel convolution. A patch is zoomed-in for better observation. **f–i** Convolutional results with different applied kernels of horizontal Sobel, vertical Sobel, blurring, and sharpening, respectively. The weights of the kernels are provided by the bar charts. **h** and **i** are zoomed-in for better observation.

## Discussion

In our experiment, the PTFP chip is operated at the speed of 20 Gbaud, corresponding to a throughput of 480 GOP/s. The computing density of the core part on-chip (electronics excluded) is 588 GOP/s/mm². With a larger scale, the computing density is capable to surpass 1 TOP/s/mm² (discussed in Supplementary note 7). Since the ODLs play a key role in the tensor processing, their insertion loss

and footprint of them are determinants of the signal-to-noise ratio, throughput and computing density of the PTFP chip. The length of ODLs is inversely proportional to the clock frequency. Advanced electrooptic modulators[37,38] and PDs[39] with large bandwidths allow higher clock frequency: the length of ODLs is thus shortened. Recent progress on ultra-low-loss silicon nitride waveguides[40] enables complicated optical delay manipulations. Based on the validation of

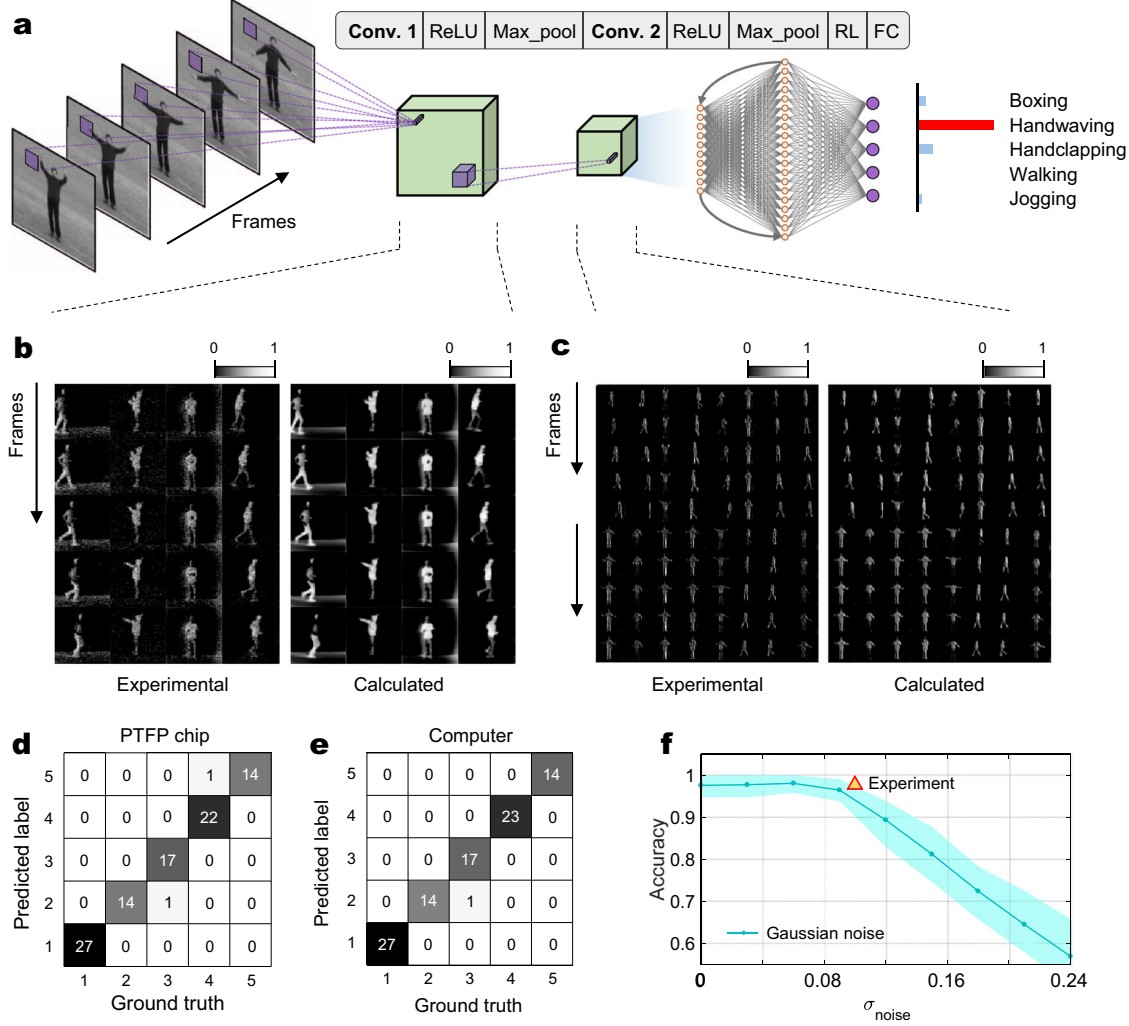

**Fig. 4 | Experimental results of video action recognition of the KTH dataset.** **a** The adopted neural network model. Input data is a segment of video with five frames. The neural network comprises two convolutional layers with activation and pooling layers, a recurrent layer (RL), and a fully connected layer (FC). The 'Conv. 1' and 'Conv. 2' layers are computed by the PTFP chip. **b** and **c** Convolutional results of the PTFP chip of Conv. 1 and Conv. 2, respectively. Subplots from top to bottom display the convolved images of different frames. From left to right, several convolved video segments are displayed. For reference, computer-calculated results are provided aside. **d**, **e** Diffusion matrices of recognition of the PTFP chip and that of a digital computer, respectively. Numbers on the diagonal line record correct prediction. **f** Simulated accuracy of the neural network with different standard deviations ($\sigma_{noise}$) of additive Gaussian noise. The solid curve represents the average recognition accuracy and the shading indicates the 90% confidence interval. Yellow triangle marks the experimental accuracy of the PTFP chip.

the key functionalities of the PTFP concept, it is feasible to improve the performance of integrated photonic devices and the completeness of high-order tensor convolution. Refining the insertion loss of the chip will contribute to a better signal-to-noise ratio. Increasing the integration scale with extra ODLs and weighting banks will upgrade the current chip to a complete four-order tensor processor (discussed in Supplementary notes 5 and 6). The upper limit of weighting bank duplication is discussed in ref. 33 and Supplementary note 6. The difference for a PTFP chip to achieve a larger kernel size is the additional introduction of waveguide crossing (virtual breaking points in Fig. 1b). Recent works show that insertion loss below 0.04 dB/crossing is obtainable[41,42], implying that the influence of waveguide crossing can be minor.

In conclusion, we propose an integrated photonic tensor flow processor which is, compared with mainstream GeMM processors, capable to process high-order tensor convolutions without extra input data transformation and memory use. The wavelength dimension carries different channels of the input tensor and the space dimension represents different channels of the output tensor. Between the input and the output, optical time delays, weighting,

and summation perform convolutional operations. The hybrid manipulation of optical wavelengths, space dimensions, and time delays offers us the opportunity to process tensors in a 'flow' fashion. The PTFP occupies less memory than the mainstream GeMM processors when the stride is smaller than the kernel width. In the proof-of-concept experiment, a silicon-based photonic chip is fabricated. It performs a two-order tensor convolution kernel shaped [1 × 3, 4, 1], demonstrating the key functionalities of the PTFP. Enlarging the integration scale will complete a four-order tensor convolution processor with currently available photonic integration technologies. A CNN is built and the video action recognition task is performed with high accuracy of 97.9%. Given the fact that a major performance bottleneck of state-of-the-art GeMM processors is the limit of memory volume, the concept of PTFP may become an effective way for high-performance processors. Enabled by photonic ultrafast clock frequency, the proposed PTFP is advantageous in parallel processing with large-batch data and is promising to promote advances in compute-intense applications such as multi-stream video processing, high-resolution surveillance, autonomous driving, and the Internet of Things.

## Methods

### Experimental setup

The experimental setup is illustrated in Supplementary Fig. 1. The input signals for the PTFP chip are generated by an arbitrary waveform generator (AWG, Keysight M8194A) with four independent output ports. Every output port works at a sampling rate of 120 GSa/s. The amplitude of the generated signals is firstly managed for proper modulation depth using RF attenuators (Rebes RBS-69-26.5-7) and amplifiers (Connphy CLN-1G18G-3025-S, bandwidth 1–18 GHz), and then modulated onto the optical carriers via electro-optic modulators (EOSPACE AX-0S5-10-PFA-PFA). Optical carriers are provided by a four-channel laser source (Alnair labs TLG-200). Erbium-doped fiber amplifiers (Ashow ASHPFA-C-23-FA-B) are used to compensate for the insertion loss of the PTFP chip. Before the optical signals enter the PTFP chip, tunable delay lines (General Photonics VDL-002-15-10-S-PP-FC/APC) are adopted to compensate for the difference in optical paths from the modulators to the chip input ports. After the output of the chip, additional tunable delay lines are used to compensate for the difference in optical paths from the chip output via PD (Lab Buddy DSC-R412) to the EPC. A real-time oscilloscope (Keysight UXR0134A) is employed for testing and recording results. The sampling rate of the oscilloscope is configured at 128 GSa/s and the obtained data is downsampled to 20 GSa/s to reconstruct the results. Note that a high sampling rate allows us to perform input and output synchronization more precisely but a sampling rate of 20 GSa/s is enough to carry out convolutions after synchronization is done. The AWG is synchronized with the trigger of the oscilloscope for steady waveform acquisition. A computer is adopted in the experiment for controlling the AWG by sending waveform files. It also performs the recurrent layer, fully connected layer, and the nonlinear process (ReLU, max pooling, and softmax) in the CNN experiment. The voltages on the chip (including WDM tuning and MRR tuning) are supplied by a homemade 45-channel voltage source, which is controlled by the computer as well. The absolute values of weights are applied via two steps: measuring the modulation curves of MRRs (shown in Fig. 2g); translating trained weights to applied voltages. In the experiment, the negative sign is applied manually by reconfiguring the experimental setup. For the multi-channel image convolution experiment, the negative sign is applied by connecting the optical signal to the negative input port of the balanced PD (Lab Buddy DSC-R412). For the CNN experiment, the chip works twice to perform positive and negative weights, respectively. Results are obtained by adding these two parts. We note that, in the final version of the PTFP system, the manual operations for negative weights should be eliminated by using both the through and drop ports of the MRRs with balanced PDs, as demonstrated in refs. 23, 33.

The heat sink inside the packaging module is controlled by a thermal controller (Thorlabs ITC4001), maintaining the chip temperature at 28 °C. At this temperature, the standard deviation of MRR weight control is 0.035 out of range 1, corresponding to 4.83 bits of resolution. The resolution is majorly limited by the MRR crosstalk. Without temperature maintenance, the resonance of MRR will significantly deviate, degrading the bit resolution. Recent approaches to accurate MRR weight controlling[43] are beneficial for the PTFP.

From a developing perspective, we present a system schematic comprising the photonic chip and the electronic periphery circuitry (shown in Supplementary Fig. 2). Compared with conventional electronic GeMM circuitry, the PTFP cancels data duplications inside the buffer. Its memory use is identical to that of the original data. Given that high-speed digital-to-analog converters (DACs) apply time multiplexing technology, their input clock rate is compatible with a conventional static random-access memory (SRAM) and their output clock rate is compatible with the photonic chip. Therefore, the high-speed DACs do not introduce obvious latency to the system. The photonic chip can operate at its peak throughput given enough memory interface width for data feeding. At the output, the analog-to-digital

converters (ADCs) apply time demultiplexing so that the output clock is compatible with the electronic memory. Overall, the speed advancement of the photonic chip can be realized provided the usage of multiplexing DACs and demultiplexing ADCs. With the detailed calculation in Supplementary note 7, we show that the high-speed DACs and ADCs will introduce an energy overhead of around 6.28 pJ/symbol. With the expansion of the integration scale, it is promising to dilute the energy overhead of DACs and ADCs down to 0.011 pJ per operation.

Although the adopted EPC is now bulky, there are two routes to enable on-chip miniaturization of the electrical signal combination. One is packaging wideband EPC dies (over 40-GHz bandwidth is commercially available) with the photonic chip using wire bonding or similar technologies. Another is to place the PDs compactly and connect their output wires directly. As long as the PDs are close enough (sub-millimeter distance), the photocurrent obeys Kirchhoff's current law so that the outputs are added without an EPC[32]. Note that the direct PD connection will accumulate their parasite capacitance and lower the final bandwidth. Therefore, this approach is feasible with high-speed PDs whose bandwidth is multiple times higher than required.

### Waveform encoding and decoding

The grayscale pixels are firstly reshaped to a row vector and the grayscale values are converted to analog waveforms by the AWG. Under the symbol rate of 20 Gbaud, the generated signal should cover the bandwidth from direct current (DC) to 10 GHz. The majority of power distributes at a low-frequency range (see Supplementary Fig. 14). However, due to the inferior loss performance of the fabricated chip (see Supplementary note 5 for details), we adopt electrical power amplifiers in our experimental setup to enhance modulation depth so that the output signal is detectable. The adopted electrical power amplifiers (nominal bandwidth: 1–18 GHz) filter out low frequencies, leaving obvious and unrecoverable distortions to the analog waveforms. To avoid such distortions, we shift the center frequency of the input waveforms from DC to the 10-GHz carrier by introducing an encoding method. Original input waveforms are multiplied by a sequence with alternating ±1:

$$Y_i = (-1)^{i+1} \cdot X_i \tag{1}$$

where $Y$ denotes the encoded waveform and $X$ is the original waveform. A waveform of a constant value becomes a sine waveform of 10 GHz. After such encoding, the majority of signal power distributes at 10 GHz, located in the bandwidth of the electrical power amplifier. Since the sampling rate of the AWG is configured at 120 GSa/s, the waveform is interpolated six times with the 'spline' method provided by the MATLAB code.

After the input waveform encoding, the convolutional kernels should be also encoded. Supplementary Fig. 15a shows the convolved result with an encoded input waveform and a non-encoded kernel ([1, 0.5]). Because of the input encoding, originally positive values are converted to negative values. As a consequence, the negative value of delayed waveform $x(t + \tau)$ is at the same position as the positive values of original waveform $x(t)$. Summing up these waveforms directly results in an error. Supplementary Fig. 15b shows a result of an encoded kernel ([1, −0.5]). In this case, the waveforms are summed up correctly. The method of kernel encoding is multiplying a mask. Each row of the mask contains alternating 1 and −1. If the length of the input image ($L$) is even, the starting sign of every row is always positive (Supplementary Fig. 15c). If $L$ is odd, the starting sign of each row alternates relevantly (Supplementary Fig. 15d).

Through the above encoding method, the center frequency of convolved results is shifted to 10 GHz, so they are decoded (multiplying the encoding sequence again) to obtain the final results. Although the majority of the signal power is preserved via encoding,

the low-frequency block property of the system still introduces some distortions to the result. Supplementary Fig. 16 illustrates an example of the result of the kernel [1, 0, −1]. It is obvious that the temporal waveform is distorted: there are many relaxation oscillations compared with the ideal result. The reason is that the low-frequency component of the waveform is filtered out. It indicates that the error of the convolved results not only comes from the experimental noise but also the waveform distortion originating from the frequency response. In other words, the error power in the experiment is the combination of noise power and distortion power. Figure 4f shows that although the standard deviation of combined error is 0.1, the recognition accuracy is a little bit higher than that of pure noise at the standard deviation of 0.1. This gives an empirical observation that waveform distortion is less impactful than noise on recognition accuracy.

Given that the encoding and decoding processes introduce data transformation overhead and undesirable signal distortion, we pursue removing the encoding and decoding in future works. Using low-drive modulators (such as the lithium niobate film (LNOI) modulators[37]) will significantly refine the modulation depth, so the electrical amplifiers and encoding/decoding process can be removed. Also, refining the loss performance of the chip benefits the output signal power. Supplementary Fig. 17 shows the effect of insertion loss and modulation depth on the output signal quality. It is found that low-drive modulators and low-loss photonic chips promise good signal quality without data transformation overheads. Although integrating LNOI modulators with silicon photonic circuits is challenging and may introduce extra loss, complexity, and optical nonlinearity, recent breakthroughs in LNOI modulators and heterogeneous integration technologies are quite inspiring for the performance refinement of the PTFP concept.

## Input and output synchronizations

In our experiment, signals are summed up in the analog domain. A symbol only lasts for 50 ps under the clock frequency of 20 GHz, so the misalignment of analog symbols gives wrong convolution results. Moreover, the adopted sampling rate is 20 GSa/s (down-sampled from 128 GSa/s), equal to the modulation rate. Only if the sampling clock and the signal generator are synchronized, correct results can be sampled. Therefore, it is necessary to conduct synchronization. The adoption of off-chip fiber devices introduces unknown optical lengths; thus, we compensate for such length variations. Tunable delays are used both for input ports and output ports, as illustrated in Supplementary Fig. 1. Before synchronization, all MRRs are set to the weight of 1. For input synchronization, two input ports and one output port are used. Two 10-GHz sine waveforms with opposite initial phases are loaded onto these ports. With a proper delay, the summed amplitude is zero. By reading the waveform of the output port, the synchronization status of the input channels can be measured. The chip has four input ports, so the input synchronization is repeated until all input ports are synchronized. Similarly, for output synchronization, one input port, and two output ports are used. 10-GHz sine waveform is input via the input port. Output synchronization is accomplished if the output waveform of the EPC reaches the highest amplitude.

## Preprocessing of the KTH dataset

The KTH dataset[35] is composed of video clips of six categories of human action ('Boxing', 'Handwaving', 'Handclapping', 'Walking', 'Jogging', and 'Running'). Each action is repeated by 25 people in four scenes. Because the recurrent layer and the fully connected layer are sensitive to image shifting, a small movement of the action subject (i.e. the person) may lead to severe overfitting. Therefore, for the static actions (boxing, handwaving, and handclapping), the video clips are cropped to put action subjects in the center of the image. This is accomplished by the following steps: (1) find the moving object (the person); (2) locate the center of the moving object; and (3) shift and crop the image. To find the moving object, we accumulate motions of different frames of the video:

$$M(x,y) = \sum_{i=1}^{N} |I(x,y,t=i) - I(x,y,t=i+1)| \qquad (2)$$

where $M(x, y)$ is the accumulated motion and $I(x, y, t = i)$ is the image at the $i$th frame. Supplementary Fig. 18a–c show three examples of motion accumulation. Find the center of the accumulated motion and the object is shifted to the image center.

For moving actions (walking, jogging, and running), if the subject moves out of the image, the CNN will also overfit the datasets. The purpose of preprocessing is to pick up frames with subjects inside the image. This is achieved by following steps: (1) check motion inside the image and (2) confirm the cause of the motion. By subtracting adjacent frames, we can find the moving part of the image. When the amplitude of motion surpasses a threshold, it is regarded as a valid motion. However, the movement of the camera (background) instead of the subject will result in valid motions, so we should confirm the cause of the motion. Observe the Fourier spectrum of the valid motion (Supplementary Fig. 18d–f). If the motion is caused by camera movement, its spectrum is white-noise-alike. Therefore, we can pick up those frames with moving subjects.

After the preprocessing of video clips, every five frames are grouped as a video segment. The Original frame rate of the KTH dataset is 25 frames per second. To enhance contrast among frames, the frame rate of the video segments is set at 12.5 frames per second. Generally speaking, recognition against 'Jogging' and 'Running' is more difficult than other categories. We use five action categories except for 'Running' for recognition. Because only when the recognition accuracy of the baseline CNN itself is high enough, we can determine whether the accuracy degradation comes from the CNN model or the PTFP chip. From the five categories, we generate 2499 video segments. These segments are flipped left to right for data augmentation. So, there are 4998 data examples for training and testing.

## CNN structure and training

The CNN for human action recognition comprises four layers: convolutional layer 1 (Conv. 1), convolutional layer 2 (Conv. 2), recurrent layer (RL), and a fully connected layer (FC)[44]. The size of convolutional kernels for Conv. 1 and Conv. 2 is $1 \times 3 \times 3$. It is a three-dimensional kernel, with the first dimension being 1. So, it can be accomplished with our experimental setup. Take the channel numbers into account. The shape of the kernel tensor of Conv. 1 and Conv. 2 is $[1 \times 3 \times 3, 1, 4]$ and $[1 \times 3 \times 3, 4, 8]$, respectively. ReLU activation function is applied after each convolutional layer. Max pooling layer is adopted to shrink the size of images with a stride of 2. The output of the second max pooling layer is a tensor of 5 frames. Each frame comprises 8 convolved feature maps. These feature maps are reshaped to five vectors as the input for the recurrent layer. Every vector is a time step in the recurrent layer. The adopted recurrent cell is the basic long–short-term memory cell provided by the TensorFlow framework. The activation function of RL is ReLU. The number of hidden neurons in a recurrent cell is 256. Finally, the FC layer transforms all hidden neurons into a recognition vector with softmax activation. The largest value in the recognition vector indicates the predicted category. To train the CNN, the dataset is separated into two parts: 3998 video segments for training and 1000 segments for testing. The loss function is defined as the cross entropy between the recognition vector and the label vector:

$$\text{Loss}^{(\Theta)} = -\sum_{i=1}^{5} y_i \cdot \log\left(\hat{y}^{(\Theta)}\right) + \gamma \sum_{j} \left(z_{\text{conv2},j}^{(\Theta)}\right)^2 \qquad (3)$$

where $y^{(\Theta)}$ is the output recognition vector under the parameter of $\Theta$. Additional L-2 regularization is adopted with a penalty coefficient of

$\gamma = 0.005$. $z^{(\Theta)}_{conv2}$ is the output of the second convolutional layer. The gradient descent optimizer is used with a learning rate of 0.01 and the network is trained for 125 epochs. The stochastic gradient descent method[45] is adopted with a mini-batch size of 50. CNN training is carried out by a computing platform with an Intel i5-9500 CPU and Nvidia RTX-2080 GPU. Supplementary Fig. 19 shows the loss functions and recognition accuracy during training. The training loss decreases consistently, indicating a successful fitting. The validation loss reaches the lowest at around 60 epochs and increases a little bit, indicating that minor overfitting occurs. It is observed that the recognition accuracy is not obviously affected by this minor overfitting, so we adopt the trained parameters at 125 epochs as the final parameters for inference.

### CNN simulation under Gaussian noise

The PTFP chip is essentially an analog processor, thus involving experimental noise and signal distortions during processing. Rich literature reveals that deep neural networks, including CNN, are partly robust to noise, especially in classification tasks[46,47]. Such empirical study indicates that we are able to achieve accurate classification with a relatively noisy analog processing system. However, the noise robustness must have a limit. We simulate the CNN with different levels of noise to find the limit of our experiment. Since the PTFP chip conducts convolutional layers, we introduce additive Gaussian noise to the output of the convolutional layers. In our experiment, 96 segments are selected for testing. So, for every noise level (standard deviation from 0 to 0.24), we randomly pick up 96 segments in the 1000 testing segments to calculate the recognition accuracy. The above procedure is repeated for 100 times to obtain the stochastics of the accuracy. Finally, the mean value and 90% confidence interval of the accuracy are shown in Fig. 4f.

## Data availability

The KTH dataset used in this study is available at https://www.csc.kth.se/cvap/actions/. The 'traffic camera' dataset used in this study is available at https://aimagelab.ing.unimore.it/visor/video_categories.asp.

## Code availability

The code for convolutional neural network training is provided by the authors in the repository: https://github.com/xsf19950411/PTFP, https://doi.org/10.5281/zenodo.7340586. Cite the current article for using the code.

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

## Acknowledgements

The authors acknowledge Prof. Qunbi Zhuge for the use of the arbitrary waveform generator (Keysight M8194A) and Mr. Shiyu Hua for helpful technical discussion. This work is partly supported by the National Key Research and Development Program of China (No. 2019YFB2203700) and the National Natural Science Foundation of China (Nos. T2225023, 62205203).

## Author contributions

S.X. and W.Z. conceptualized this study. S.X., J.W., and S.Y. came up with the methods. J.W. conducted the simulation and designed the chip layout. S.X., J.W., and S.Y. carried out experiments and obtained data. S.X. and J.W. visualized the data. W.Z. supervised the work. S.X. and J.W. wrote the original manuscript. S.X., J.W., and W.Z. revised and edited the manuscript. S.X. and W.Z. funded this study.

## Competing interests

The authors declare no competing interests.
