## [Peer Review File · Nature Communications]

High-order tensor flow processing using integrated photonic circuitsREVIEWER COMMENTS

Reviewer #1 (Remarks to the Author):

The authors have presented a photonic integrated circuit that is specifically designed to perform linear operations (weights) on a WDM analog data stream. This represents a noteworthy evolution from their earlier theoretical work on using the degrees of freedom of light to facilitate linear operations by photonic devices (Refs 32, 33). This work's stated claims are significant, offering an avenue for high performance tensor processing that are very useful for machine learning algorithms. The article provides a set of references of similar works in the established literature (refs 21-31), and states that the main innovation is the "hybrid use of multiple degrees of freedom of light to process high-order tensor instead of GeMM factorization". The motivation for avoiding the GeMM is to avoid memory inefficiency due to digital data "duplication and shifting", and doing the processing instead via a "flow" processor, without requiring significant memory resources.

To support this claim, the authors provide an optical circuit design and a small-scale experimental demonstration that is able to partially perform tensor products, ubiquitous in convolutional neural networks. Though the presented results narrowly qualify as a promising avenue towards a tensor processor, it does not support the claim resulting from the motivation for the work, namely the lack of data duplication and complexity. It also does not support the claim of high speed (20GBaud, not a proper unit for speed for analog tensor processing) since data was loaded in an offline manner, via an AWG, and not in real time. The statement regarding accomplishing 20GHz operation in the abstract is therefore plainly false. Furthermore, the paper does not address the potential toward this speed since it does not take into account the large overhead in the data preparation at the input and detection at the output. Such overhead, unless proven otherwise, would negate any benefits coming from this photonic chip relative to the state of the art alternatives. As a result, I recommend the editor not to accept this manuscript in its current form. In my opinion, it needs major revisions to the evidence and a readjustment of the claims -- in other words, a resubmission.

My major concerns are enumerated as follows.

- On page 2, the authors claim the GeMM is undesirable because it involves duplication and shifting (many times) incurring memory access penalty. The PTFP would directly process high-order tensors without transformation. In the rest of the article, however, they contradict this statement by using many examples of data transformation to map inputs to their chip. First, a very expensive DAC conversion with data reshaping and duplication to create a 3x3 kernel (all done by an AWG). Second, temporal modulation also require memory buffering, which become extremely costly at 20GBaud. Third, a waveform/kernel encoding and decoding scheme that are not compatible with the rest of the computation "flow". Though it is claimed that it wouldn't be necessary if a low-drive linear modulator is used, no numerical simulation is provided. Fourth, regardless of encoding and decoding, both inputs and outputs are digital streams and must be converted between digital/electric to analog/optical signals between PTFPs. The authors do not present how that can be accomplished at 20GBaud without exceeding power and latency budgets of state of the art electronic tensor processors such as Google's TPU.
- I disagree with the statement on page 7 which states that the "hybrid manipulation of optical dimensions of wavelength, time, and space offers us the opportunity to process tensor in a 'flow' or 'non-stop' fashion." The very act of using time as a dimension means we must perform 'blocking' operations to serialize and deserialize time series into input/output tensors. If by time the authors refer to the delay lines present on chip, I think that it is simply a finite impulse response filter, which doesn't add a degree of freedom other than space would. I think the 'time' dimension should be removed in this particular statement about flow computation.
- It is unclear to me how negative weights can be applied. Page 5 refers to applied weights of [-1,0,1] but I am having a hard time picturing how -1 can be applied by simply examining the circuit diagram. Analog values are encoded in power amplitude of light (positive), and physical weights are set by MRR's transmission curve (from 0 to 1). The authors should clarify how this can be done. I suggest just providing an example of how two identical AC signals at different wavelengths can be subtracted from one another optically resulting in a zero electrical readout at the output.

- On a few occasions, the authors have mentioned that expanding the space dimension (by duplicating the circuit with mode splitters) is straightforward (page 4 and 6). However, I would like to note that increasing that number also requires increasing total optical power input. This has two deleterious effects: increased static power consumption, and potential distortion by nonlinear effects on waveguides and especially MRRs. Because of that, and because of footprint overall, the system cannot be scaled indefinitely. I suggest the authors provide a scalability analysis, taking into account total power consumption and nonlinear effects.
- Finally, on page 7, the authors propose this "PTFP concept is able to promote the advances of compute-intense applications such as video processing, high-resolution surveillance etc." In such applications, the involved data rates are far below "photonic ultrafast clock frequencies". How do we reconcile the >1000x disparity in speeds? Please comment on the circuitry performing this upconversion.

Minor concerns include:

- The article has quite a few grammar errors that need revision before publication.
- GBaud/s is the wrong unit. It should read Gbaud.
- Comment on bit resolution of analog representation. Only a basic simulation done at the output attempted to study the effect of low-precision analog representation, or noise overall, on the recognition accuracy. Can you comment on the control challenges of setting weights with MRRs and its sensitivity to crosstalk and environmental fluctuations?

Reviewer #2 (Remarks to the Author):

In the article, "High-order tensor flow processing using integrated photonic circuits," the authors present their work which uses temporal and frequency space to achieve convolutional operations with low latency. Overall the authors have demonstrated an impressive work that uses time delay lines (the main novelty of this manuscript) to enable summation using the time domain. The authors have introduced the topic well and have cited the relevant literature in the introduction. I also have to applaud the authors for the very detailed supplementary information which contains the information needed to replicate their results. I support publication in Nature Communications after the authors address the following concerns:

1. The main issue I have with this work is the challenge in understanding how time, frequency, and MRRs all map to the input features and convolutional kernels. The authors have the tensors represented in abstract diagrams in figure 1. It would be very helpful to the reader if the authors could instead represent these tensors as actual matrices like in some of the papers they cite (e.g., similar to Fig 1c in Feldmann et al., Nature 589, 52-58 (2021)). Even now reading the authors' manuscript, I find it very challenging to understand how the time dimension was used. For example, I'm not sure how the authors are implementing the 3x3 kernel shown in Fig. 3g since this cannot be simplified to a single 1x3 kernel used 3 times (like in 3d,e, and f).
2. Since the authors are delaying the optical signal from coherent laser sources, they are forced to use high speed electronics ("EPCs") to recombining the signals electrically to prevent unwanted optical interference. This could be a limitation of this approach as RF power combiners can be bulky and limited in bandwidth (esp. flatness in amplitude and phase). The authors should discuss this and possible routes to address this in on-chip integration.
3. The authors mentioned duplication of data and memory access bottlenecks when performing convolutions on other platforms. This can be addressed by better pipelining of weights and input data (for example, the Caffeine optimizer demonstrated in 2016, DOI:10.1109/TCAD.2017.2785257). The authors should discuss more up-to-date methods for implementing CNN in hardware so as not to construct an overly simplistic view of how digital hardware actually performs such operations (for example, full "memory access" for each time data is reused is not the case).
4. If I understand correctly, the main use of the time delay is for convolutions with a stride of 1 where

the input data is duplicated. If that is the case, convolutions which have a stride equal to the width of the kernel do not have data duplication (like Supplementary Fig. 19). I don't believe this was mentioned or motivated in the main text, so the authors should discuss the specific cases where their time delay method is useful for CNNs.

Reviewer #3 (Remarks to the Author):

The authors of this manuscript claim the demonstration of an integrated photonic tensor processor that performs operations on higher dimensional tensors without the need for conversions to matrices beforehand. This is presented as the key novelty of the paper. They claim to exploit various degrees of freedom in optics such as wavelength, time and space to do so and report on 20 GHz operation which is used in an image sequence classification benchmark, yielding 97.9 % accuracy. This is done with the photonic circuit performing convolution operations as part of a larger CNN implemented with conventional computer hardware. Furthermore, the authors claim comparable performance of the photonic solution to electronic counterparts with potential to even surpass that.

In my opinion, most of the claims are supported by the provided data and evidence, albeit some claims are formulated in a too strong manner and need to be revised. More detailed comments about that will be given later in this report. The main experimental results seem to be obtained carefully with the methodology outlined in detail both in the main manuscript and in the supplementary material. Various questions are still left open which will be described later in this report and should be addressed by the authors. The experimental data and evidence is presented in a structured and consistent way, the supplementary material is well organized and very complete.

The main issue that needs clarification is the claim of the photonic processor not relying on memory access and tensor to matrix conversions in comparison to other work out there. However, throughout the methodology section, it becomes clear that several data conversion steps (e.g. 2D image to 1D vector conversion among many others) are still needed. In addition, the size of the kernel in experiment is 1 in the first dimension which seems ill-chosen given the major point of the solution to be able to deal with higher-order tensor processing directly in the photonic chip. A 2D kernel operation is described on page 5, 2nd paragraph, and seems to involve multiple iterations of the hardware. It would have been a much more convincing result if the spatial part of the chip is extended to deal directly with 2D kernels. At least, that would be expected from the way the claim is formulated. Furthermore, memory is in my opinion still used during the operation to set the MRR banks to the desired kernel values.

Significance of the work

The presented work is of high significance as to the best of my knowledge it demonstrates parallel processing of tensors reflecting multiple input channels by a photonic circuit. Although there is a discussion to be had on the dimensions of the chosen data per input channel, the results show that the photonic solution can deal with multiple channels at the same time through wavelength multiplexing, which provides an advantage over matrix multiplication hardware as such. Within each input channel the high speed of the photonic solution is exploited to be able to deal with the information in a serial way.

Data and methodology

Extensive experimental data is provided for the subcircuits of the photonic chip and supported by simulations. Furthermore, data for validating the convolution operation and demonstrating it in a benchmark is given. Points as listed below should be addressed by the authors however:

(1) The current data to proof the convolution operation is given in Fig. 3c. However, the input data is not provided and it is difficult to judge the correctness of the operation. Indeed, the calculated result is

overlayed and it has been made clear that the experimental clock rate leads to a sampling of the ideal data. However, more discussion on the conditions of this sampling is needed. Is the sampling sufficient and how close should the experiment be to calculated result so that it is sufficient? Are we operating at the limit and would a higher sampling rate improve performance?

(2) In Fig. 3d-g the input image is not shown, which would give better means to compare the output from the convolution. Why were different input images chosen for this validation? Please explain the reasoning behind.

(3) It seems that the results 3d-g are obtained by exploiting different channels to yield a 2D 3x3 kernel. This relates to the main issue mentioned above. Given the claim, one expects different input channels and 2D kernels simultaneously and not used to compensate for each other. Please revise the claim in a way that better reflects the experimental limitations here.

(3) In page 6, supplementary material, the authors point to meter long delay lines for larger kernels as those scale with kernel size. It is hinted that those can be implemented using fiber. Given the direction of integration and argumentation of high throughput density, anything off chip would be counter intuitiv. Please address the issue of scalability of the kernel and connected with that the limitations to chip area, throughput density, due to delay lines. Also link that with the author's discussion on scaling S7. Furthermore, it is difficult to understand how 'advanced electrooptic modulators, PDs, and electrooptic packaging technologies are beneficial to shortening on-chip ODLs'. What is the relation here?

(4) In S7, the throughput is calculated. I don't understand the argument of 2 operations per clock. Given an information coding of 20 GBaud, that means 20E9 values of pixels per second which would also yield 20E9 operations per second on those pixels. I understand the multiplication with the kernel of 12 weights. Please address this.

(5) On page 8 of the manuscript, the input coding is explained that is used to circumvent the bandpass characteristic of the used amplifiers. Given that the sign is changed between each consecutive bits, this means that only one sample exists per symbol. I assume that after the analog modulation, the analog signal stays at those values for the duration of the bit period. How does the sampling affect the processing of the signal. Are we operating at the limit of nyquist sampling here?

(6) In same section, the authors write 'Since the error of waveform distortion is relevant to the waveform itself, its influence on the final recognition accuracy may be less than the experimental noise'. This is not clear at all. Are there two aspects that now distort the signal, one being frequencies <10 Ghz not being passed by the amplifier and the second being the coding scheme not having signal components below 0.5 GHz as Fig. S15 shows? How do each of those aspects influence the recognition of the initial images?

(7) Please explain in more detail the reasons to apply cropping and pre-processing to the KTH dataset. It is not clear why that is needed.

(8) On page 10 the authors state that overfitting does not affect the recognition accuracy. I think this is not true. Please revise that part. Data shown in Fig. S17 clearly indicate overfitting. As such it is maybe only a problem for the computer baseline and not relevant to its application on the photonic solution.

(9) When applying weights to the photonic chip, what was the strategy followed here? Could the authors detail this more? The training will yield a model that is optimized for the digital computer. As the photonic chip is analog, one might expect certain translation might be needed for the weights.

(10) Fig. 4f shows the simulated performance in presence of Gaussian noise. Where is the noise added to in specific? Is that to the data signals or weights? Please provide more details here.

Analytical approach

The authors use a simple convolution test to validate the photonic hardware and employ the KTH benchmark to test its real operation. Both methods seem adequate and suitable to be used here. Furthermore, the authors analyse the influence of noise which is important given the analog noisy nature of a photonic solution.

Improvements

A few aspects as outlined below would improve the understanding of the work.

(11) Could the authors discuss in more detail how many wavelengths can be used in parallel and thus how the amount of input channels can be scaled?

(12) Energy-efficiency is mentioned to be one strong point of a photonic solution. Could the authors include discussion of the energy consumption of the existing solution? So far, no discussion on energy is given and no metrics calculated.

(13) The caption in Fig. S8 is wrong and not related to MRRs.

(14) The chip insertion loss is given to be around 30 dB. This is rather high. Could the authors comment on the effect of loss on the actual experiment? In S6, the influence of loss on SNR is discussed but more on a theoretical level. Was the loss a problem in experiment? And how to get to 7.4 dB optical link loss for sufficient SNR as discussed in S6?

Clarity and context

The authors clearly describe the motivation, conceptual framework followed, photonic hardware and experimental results. A few areas could be improved in terms of clarity, as listed below:

(15) The MRR weighting characterization as shown in Fig. 2g is not clear. Why is there a difference between the rings and what is desired?

(16) The authors write that the proposed solution allows the signals to 'flow' and perform the needed operation. This concept of flowing is very vague and not clearly defined. Please revise that in the respective sections.

(17) Fig. 4a depicts the general CNN architecture for the benchmark experiment. Could the authors add the pooling layers and feature maps etc and indicate which parts are performed in photonics and which parts in conventional computer?

(18) In the conclusion, the authors state that the photonic solution demonstrates processing with a four-order kernel. Could the authors explain that in more detail to me?

Expertise

I would like to disclose here the lack of expertise with respect to convolutional neural networks and their in-depth operation to be able to fully assess the methodology and results related to Fig. 4. Furthermore, I am not familiar with the KTH dataset.

Thank you very much for the constructive comments. Your comments greatly improve the quality of our work and manuscript preparation. The responses to all the concerns are described one by one as follows:

To Reviewer #1:

Black: Comments from the reviewer.

Blue: Response of the authors.

Red underlined: Revisions to the manuscript.

The authors have presented a photonic integrated circuit that is specifically designed to perform linear operations (weights) on a WDM analog data stream. This represents a noteworthy evolution from their earlier theoretical work on using the degrees of freedom of light to facilitate linear operations by photonic devices (Refs 32, 33). This work's stated claims are significant, offering an avenue for high performance tensor processing that are very useful for machine learning algorithms. The article provides a set of references of similar works in the established literature (refs 21-31), and states that the main innovation is the "hybrid use of multiple degrees of freedom of light to process high-order tensor instead of GeMM factorization". The motivation for avoiding the GeMM is to avoid memory inefficiency due to digital data "duplication and shifting", and doing the processing instead via a "flow" processor, without requiring significant memory resources.

To support this claim, the authors provide an optical circuit design and a small-scale experimental demonstration that is able to partially perform tensor products, ubiquitous in convolutional neural networks. Though the presented results narrowly qualify as a promising avenue towards a tensor processor, it does not support the claim resulting from the motivation for the work, namely the lack of data duplication and complexity. It also does not support the claim of high speed (20GBaud, not a proper unit for speed for analog tensor processing) since data was loaded in an offline manner, via an AWG, and not in real time. The statement regarding accomplishing 20GHz operation in the abstract is therefore plainly false. Furthermore, the paper does not address the potential toward this speed since it does not take into account the large overhead in the data preparation at the input and detection at the output. Such overhead, unless proven otherwise, would negate any benefits coming from this photonic chip relative to the state of the art alternatives. As a result, I recommend the editor not to accept this manuscript in its current form. In my opinion, it needs major revisions to the evidence and a readjustment of the claims -- in other words, a resubmission.

Response:

We greatly appreciate your detailed review and professional comments. Your comments help

improve the quality of our work and the manuscript. We are happy to address your concerns one by one in the following sections.

Comment 1:

My major concerns are enumerated as follows.

On page 2, the authors claim the GeMM is undesirable because it involves duplication and shifting (many times) incurring memory access penalty. The PTFP would directly process high-order tensors without transformation. In the rest of the article, however, they contradict this statement by using many examples of data transformation to map inputs to their chip. First, a very expensive DAC conversion with data reshaping and duplication to create a 3x3 kernel (all done by an AWG). Second, temporal modulation also require memory buffering, which become extremely costly at 20GBaud. Third, a waveform/kernel encoding and decoding scheme that are not compatible with the rest of the computation "flow". Though it is claimed that it wouldn't be necessary if a low-drive linear modulator is used, no numerical simulation is provided. Fourth, regardless of encoding and decoding, both inputs and outputs are digital streams and must be converted between digital/electric to analog/optical signals between PTFPs. The authors do not present how that can be accomplished at 20GBaud without exceeding power and latency budgets of state of the art electronic tensor processors such as Google's TPU.

Response:

Thanks again for pointing out the insufficiency of our previous manuscript. We would like to address the issues mentioned in *Comment 1* from four aspects.

As a summary, most redundant data transformations (such as data duplication, encoding, and decoding) in the AWG are caused by experimental compromises under limited conditions in our lab. These transformations are not needed for future development. As a research work, this manuscript aims at demonstrating the feasibility of the PTFP idea via a proof-of-concept experiment. However, according to your comments, we are aware of the problems in our previous manuscript: the viable routes for development are not deeply studied and discussed; some descriptions are imprecise; some claims are stated in a too strong manner. So, we are happy to revise our manuscript and address your concerns.

(1) It is appreciated for the reminder about the precision of claims. Regarding the 3x3 kernel example, we think that we should clarify the exact functionality of the PTFP, instead of simply claiming that it 'does not need data duplication'.

In principle, the PTFP can reduce data duplication for N times (N being the number of photonic time delays). The fabricated chip contains 3 delay steps so it can conduct 1x3 kernels without data duplication (demonstrated in Figs. 3c and 3d). In the experiment, to validate the capability of multi-channel convolution, we use three wavelengths to conduct three parallel 1x3

kernels to an identical image so that an equivalent 3×3 kernel is formed. Due to this process, the input image is duplicated 3 times. As a comparison, GeMM should duplicate input images 9 times to realize a 3×3 kernel.

To address the data duplication issue clearly in the manuscript, revisions are done as follows. (Lines 36-37, Page 3): “In other words, the PTFP approach saves D_{kernel} times of memory for input tensor transformation.”

(Lines 18-31, Page 5): “As we have multiple channels for input, we validate the multi-channel convolution in this part.... In the case that three input channels are configured with an identical image with row shifting, the multi-channel convolution can equivalently perform a 3×3 convolution. We should note that the current PTFP chip with 3 delay steps is not able to process 3×3 convolution directly without data duplication. However, the equivalent 3×3 convolution is a good way to benchmark the multi-channel convolution of this chip so several 3×3 convolutions are demonstrated in Figs. 3e-3i...”

(Fig. 1): Fig. 1a is also revised to better show how data is transformed with the PTFP so that readers can more easily tell the exact functionality of the chip.

Fig. 1 before:

Fig. 1 revised:

However, we would like to note that, in principle, 3×3 convolutions can be directly realized on-chip with a developed number of delay steps. The input data duplication is then not required for the PTFP. The engineering route is discussed in Suppl. notes S6 and S7.

(Lines 15-18, Page 6, Suppl. information): “Currently, the fabricated chip comprises 3 delay steps. By augmenting the number of delay steps, the PTFP can process larger convolutional kernels without data duplication. One way is to cascade more ODLs with the same length. It can enlarge the kernel size of one-dimensional convolution. The other is to introduce long ODLs to form higher-dimensional convolutions...”

(Lines 8-9, Page 8, Suppl. information): “Specifically, the area occupied by the optical delay lines is 6.25 mm^2 , 15.08 mm^2 , 24.55 mm^2 for 3×3 , 5×5 , and 7×7 kernels, respectively. It is compatible with off-the-shelf photonic integration.”

(2) Regarding the temporal modulation, we agree that it is important to discuss the memory buffering of high-speed DACs and the circuitry in practical engineering.

Referred to [B. J. Shastri, et al, Nature Photon., 2021, doi: 10.1038/s41566-020-00754-y; C. Demirkiran, et al, arXiv, 2021, doi:10.48550/arXiv.2109.01126], we provide an additional figure (Suppl. Fig. 2, attached below) to compare the envisioned periphery circuitry of the PTFP and the GeMM for a better discussion of this issue. We take Google’s TPU as a representative GeMM processor. Before sending data to digital matrix multiplication, input images are

duplicated in the buffer or the DRAM, occupying more memory resources. In contrast, the PTFP does not require data duplication in the buffer. The signals are generated in their original vector format. Besides, typical high-speed DACs generate signals via multiple stages of digital or analog time multiplexers (i.e. parallel slow-clocked digital input are converted to serial fast-clocked analog signals). Such conversion does not require extra memory resources. Therefore, the overall memory consumption of the PTFP equals the input data.

To address this issue clearly in the manuscript, we revised the Methods part as follows.

(Lines 34-41, Page 8 to Lines 1-3, Page 9): “From a developing perspective, we present a system schematic comprising the photonic chip and the electronic periphery circuitry (shown in Suppl. Fig. 2). Compared with conventional electronic GeMM circuitry, the PTFP cancels data duplications inside the buffer. Its memory use is identical to that of the original data. Given that high-speed digital-to-analog converters (DACs) apply time multiplexing technology, their input clock rate is compatible with conventional static random-access memory (SRAM) and their output clock rate is compatible with the photonic chip. Therefore, the high-speed DACs do not introduce obvious latency to the system. The photonic chip can operate at its peak throughput given enough memory interface width for data feeding. At the output, the analog-to-digital converters (ADCs) apply time demultiplexing so that the output clock is compatible with the electronic memory. Overall, the speed advancement of the photonic chip can be realized provided the usage of multiplexing DACs and demultiplexing ADCs.”

(Suppl. Fig. 2):

(Caption of Suppl. Fig. 2): “Suppl. Fig. 2 Circuitry of conventional digital GeMM and the PTFP. a, Brief circuitry of Google’s tensor processing unit (TPU), extracted from ref. [S4]. To deal with GeMM, the input data should be firstly duplicated in the DRAM or the I/O buffer (typically SRAM). Then the buffered data enters the MAC cores for matrix computing. Weights are loaded to the MAC cores via a weights buffer. b, The envisioned circuitry of the PTFP. The input data is moved to the I/O buffer and then converted to analog signals through the DAC. Since data duplication is not required, the throughput of data movement is only limited by the memory interface width. The DACs work in a time multiplexing, converting parallel low-speed electronic signals to serial high-speed optical signals. At the output of the photonic chip, ADCs work in a time demultiplexing manner, converting serial high-speed outputs to parallel low-speed digital data for storage.”

(3) Regarding the encoding and decoding methods that we applied in the experiment, they are indeed redundant and undesired in future engineering. Therefore, we include an additional statement on the reason for applying encoding and decoding methods. Also, following your suggestion, we provide a simulation result about the reason why adoption of low-drive modulators can remove the encoding and decoding.

Given the insertion loss of the current chip is measured at around 30 dB. The output signal is not detectable without amplification. Therefore, electrical amplifiers are used to enhance the modulation depth for better signal power. However, the adopted electrical amplifiers are DC-blocking (nominated passband: 1GHz-18GHz), which forces us to come up with the encoding and decoding methods. If low-drive modulators (such as lithium niobate film modulators) are available, good modulation depth is achievable directly with DACs at CMOS-compatible voltages. The encoding and decoding are then not required. The improvement in insertion loss will also facilitate the cancellation of encoding and decoding.

The effect of modulation depth (i.e. V_{signal}/V_{π}) on output signal quality (SINAD) is simulated. The result is supplemented as Suppl. Fig. S17 (also attached below). In order to reach a good signal quality, sufficient modulation depth should be applied, especially when the chip's insertion loss is relatively high. Therefore, using low-drive modulators and refining insertion loss of the chip are helpful for signal quality. The electrical amplifiers and the encoding/decoding methods are then not required in engineering.

This issue is addressed in the manuscript with the following revisions.

(Lines 14-16, Page 9): “However, due to the inferior loss performance of the fabricated chip (see Suppl. note S5 for details), we adopt electrical power amplifiers in our experimental setup to enhance modulation depth so that the output signal is detectable.”

(Lines 10-17, Page 10): “Given that the encoding and decoding processes introduce data transformation overhead and undesirable signal distortion, we pursue removing the encoding and decoding in future works. Using low-drive modulators (such as the lithium niobate film modulators [37]) will significantly refine the modulation depth, so the electrical amplifiers and encoding/decoding process can be removed. Also, refining the loss performance of the chip benefits the output signal power. Suppl. Fig. S17 shows the effect of insertion loss and modulation depth on the output signal quality. It is found that low-drive modulators and low-loss photonic chips promise good signal quality without data transformation overheads.”

(Suppl. Fig. S17):

(Caption of Suppl. Fig. 17): “Suppl. Fig. 17 Signal quality (SINAD) with different modulation depth and insertion loss. The result is simulated with identical conditions in the experiment by assuming different insertion losses. ‘IL=30 dB’ represents the performance of the current photonic chip. The output signal quality can be refined by both higher modulation depth and better loss performance. In the cases that IL is low, too large modulation depth leads to a signal quality degradation because of the nonlinearity of electro-optic modulation.”

(4) Regarding the power budget, we would like to address this issue by referring to other studies since it is a general issue for photonic analog computing.

Compared with digital processors (e.g. Google’s TPU), photonic analog processors require extra analog/digital and electrical/optical interconversion. However, there is a common agreement that, the excess power budget can be diluted with large-scale integration. In specific, referred to [C. Demirkiran, et al, arXiv, 2021, doi:10.48550/arXiv.2109.01126], the aggregate power budget for A/D and E/O interconversions is around 7 pJ/symbol. Assuming K parallel operations are conducted with one input symbol, the energy consumed for a single operation is diluted to $7/K$ pJ and the energy efficiency of photonic analog processors may surpass their electronic counterparts.

On the latency aspect, Suppl. Fig. 2 shows that, although the clock rate of photonic processors is much higher than electronics, the multiplexing DACs reconcile the clock rate difference and convert digital data into the analog domain in real-time (a minor latency <1 ns). Therefore, high-speed DACs may not introduce obvious latency to the system.

These issues are addressed in the revised manuscript as follows.

(Lines 16-21, Page 7, Suppl. information): “The analog photonic processing requires digital/analog interconversions and electrical/photonic interconversions. Compared with digital processors, the usage of DACs, ADCs, lasers, modulators, and photodiodes will consume extra power. As the number of integrated computing unit (MRR in this work) increases, the energy consumed by the interconversions will be diluted. Therefore, the energy efficiency of photonics has the potential to perform superior to its electronic counterparts with large-scale integration

[25, 27, 28].”

(Lines 37-41, Page 8 to Lines 1-2, Page 9): “Given that high-speed digital-to-analog converters (DACs) apply time multiplexing technology, their input clock rate is compatible with conventional static random-access memory (SRAM) and their output clock rate is compatible with the photonic chip. Therefore, the high-speed DACs do not introduce obvious latency to the system. ... At the output, the analog-to-digital converters (ADCs) apply time demultiplexing so that the output clock is compatible with the electronic memory.”

Finally, we believe that it is reasonable for an original article to demonstrate the feasibility of a novel idea under compromised experimental conditions. The developing and engineering works will follow up unless it is technically unrealistic. In the revised manuscript, we provide technically-sound routes for future engineering from four aspects. We hope these revisions can address your major concern properly.

Comment 2:

I disagree with the statement on page 7 which states that the "hybrid manipulation of optical dimensions of wavelength, time, and space offers us the opportunity to process tensor in a ‘flow’ or ‘non-stop’ fashion." The very act of using time as a dimension means we must perform 'blocking' operations to serialize and deserialize time series into input/output tensors. If by time the authors refer to the delay lines present on chip, I think that it is simply a finite impulse response filter, which doesn't add a degree of freedom other than space would. I think the 'time' dimension should be removed in this particular statement about flow computation.

Response:

Thanks a lot for your suggestion. We agree that we should clarify the optical dimensions used.

Indeed, the term ‘time dimension’ in the previous manuscript means the delays on-chip. Provided that the term ‘time dimension’ is strictly defined, we modified the statement about flow computation. In the revised manuscript, we refer to the delay lines with the term ‘delay steps’ rather than ‘time dimension’.

Revisions done to the manuscript are enumerated as follows.

(Abstract, Page 1): “The hybrid manipulation of optical wavelengths, space dimensions, and time delay steps, enables the direct representation and processing of high-order tensors in the optical domain.”

(Lines 3-5, Page 4): “Then, directional couplers and optical delay lines are deployed to provide the time delay steps, D_t . In each delay step, optical sequences are further split to provide the dimension of space, D_s . In a specific delay step and space dimension,...”

(Lines 2-3, Page 7): “... cascading more ODLs can achieve a larger number of delay steps: ...”

(Lines 20-22, Page 7): “The hybrid manipulation of optical wavelengths, space dimensions, and

time delays offers us the opportunity to process tensor in a ‘flow’ fashion.”

Comment 3:

It is unclear to me how negative weights can be applied. Page 5 refers to applied weights of [-1,0,1] but I am having a hard time picturing how -1 can be applied by simply examining the circuit diagram. Analog values are encoded in power amplitude of light (positive), and physical weights are set by MRR's transmission curve (from 0 to 1). The authors should clarify how this can be done. I suggest just providing an example of how two identical AC signals at different wavelengths can be subtracted from one another optically resulting in a zero electrical readout at the output.

Response:

Your understanding is correct. In principle, negative weights cannot be loaded with the positive optical intensity and positive transmission rate of MRRs.

In our experiment, the negative coefficient is equivalently realized with manual operations. For signals on different wavelengths, the negative sign is added by imposing a 180-degree phase shifting to the 10-GHz carrier during encoding. For your reference, we provide an experimental result of optically subtracting an identical waveform on two wavelengths using this approach.

For signals on different delay steps, the negative sign is added at the photodetection. Balanced photodetectors (Lab Buddy DSC-R412) are used. The optical signal is connected to the negative input port of the balanced detector to perform negative weights. This is the way we obtain Fig. 3c.

As we have declared in the response to your “*Comment 1*”, the encoding and decoding method is undesirable in future engineering, so the approach for negative weights should also be modified. Using both through/drop ports of MRRs and balanced detectors [A. Tait, et al, Scientific Reports, 2017, doi:10.1038/s41598-017-07754-z] can physically apply negative weights without manual operations.

This issue is addressed in the manuscript as follows.

(Lines 20-27, Page 8): “The negative sign is applied equivalently by manually changing the experimental setup. For signals on different wavelengths, the negative sign is applied by imposing a 180° phase shift to the 10-GHz carrier during encoding (encoding is described in the next section of Methods). For signals on different delay steps, the negative sign is applied by connecting the optical signal to the negative input port of the balanced detector. For the CNN experiment, the chip works twice to perform positive and negative weights, respectively. Results are obtained by adding these two parts. We should note that the manual operations for negative weights can be eliminated by using both through and drop ports of the MRRs as demonstrated in [23].”

Comment 4:

On a few occasions, the authors have mentioned that expanding the space dimension (by duplicating the circuit with mode splitters) is straightforward (page 4 and 6). However, I would like to note that increasing that number also requires increasing total optical power input. This has two deleterious effects: increased static power consumption, and potential distortion by nonlinear effects on waveguides and especially MRRs. Because of that, and because of footprint overall, the system cannot be scaled indefinitely. I suggest the authors provide a scalability analysis, taking into account total power consumption and nonlinear effects.

Response:

You are greatly appreciated for your kind suggestion and professional insights. There is truly an upper limit for spatial expansion because of nonlinear effects, noise figure, and footprint.

The influence of these effects and the upper limit for space expansion is theoretically studied in our previous work (S. Xu, et al, IEEE Photon. Technol. Lett., 2021, doi:10.1109/LPT.2020.3045478). In brief conclusion, the nonlinearity of the waveguide after the WDM is the bottleneck of total optical power. By using waveguides with a low nonlinear coefficient (e.g. Si₃N₄), the affordable optical power can be optimized to around 20 dBm. The scale of space expansion is also limited by the noise of amplified photodetection. Too many duplications lead to undetectable optical power at photodetectors. In specific, 32 copies of 3x3 kernel are affordable with Si₃N₄ waveguides. In this case, the total power consumption of the system is evaluated at around 10 W (majorly consumed by lasers) and the footprint of the PTFP chip is around 300 mm².

Revisions are done to address your concern.

(Lines 3-4, Page 7): “...and simply duplicating the same structure can provide additional space dimensions. The upper limit of space duplication is discussed in [33] and Suppl. note S6.”

(Lines 2-14, Page 6, Suppl. information): “The limit of the space dimension is the signal-to-noise ratio (SNR) and the factors behind SNR. Since the optical signals of all space dimensions come from the WDM, the optical intensity is limited by the nonlinear effects of the output

waveguide of the WDM. With limited optical power, increasing spatial duplication leads to low optical power in each copy. Given that the noise of amplified photodetectors is partly independent of the input optical power, the SNR is lowered with a larger spatial dimension. In [33], the maximum spatial splitting is evaluated. Using Si3N4 waveguides, when the insertion loss (additional loss excluding the theoretical splitting loss) of the optical link is less than 7.4 dB, the space dimension can reach 32 for 3×3 kernels. The optical SNR is maintained higher than 10 dB. Optical SNR higher than 10 dB means the SNR of the converted electrical signal is higher than 20 dB, which is sufficient for most classification tasks. Note that the evaluated photodetector in [33] is commercially available ones. When advanced photodetectors such as high-speed avalanche PDs are adopted, the SNR can be increased further.”

(Lines 13-15, Page 8, Suppl. information): “Among all examples listed in Suppl. Table 5, the largest chip footprint is $(24.55+64\times 7\times 7\times 6\times 0.0144)=295.5$ mm², compatible with current photonic integration technologies.”

Comment 5:

Finally, on page 7, the authors propose this "PTFP concept is able to promote the advances of compute-intense applications such as video processing, high-resolution surveillance etc." In such applications, the involved datarates are far below "photonic ultrafast clock frequencies". How do we reconcile the >1000x disparity in speeds? Please comment on the circuitry performing this upconversion.

Response:

As you suggested, we additionally describe the circuitry of reconciling clock frequencies between electronics and photonics.

The circuitry of the PTFP electro-optic system is provided in Suppl. Fig. 2, showing that high-speed multiplexing DACs act as a key role in clock frequency reconciliation, i.e. parallel digital signals are converted to high-speed serial analog signals via multiplexing. We also inspect an application scenario of virtual reality (VR) where the video format is ‘8K,120 fps’. The raw data stream is ~4 GB/s. For a server that processes multiple streams of VR videos, the ultrafast photonic clock frequency is in need. In high-resolution surveillance (e.g. traffic, security, and intelligent city), multiple streams of high-definition videos need to be processed in parallel. The photonic solution is also in need.

The manuscript is revised as follows to address this issue.

(Lines 31-33, Page 7): “Therefore, the proposed PTFP concept is able to promote the advances in compute-intense applications such as multi-stream video processing, high-resolution surveillance, autonomous driving, and the Internet of Things.”

(Lines 37-41, Page 8 to Lines 1-3, Page 9): “Given that high-speed digital-to-analog converters (DACs) apply time multiplexing technology, their input clock rate is compatible with

conventional static random-access memory (SRAM) and their output clock rate is compatible with the photonic chip. Therefore, the high-speed DACs do not introduce obvious latency to the system. The photonic chip can operate at its peak throughput given enough memory interface width for data feeding. At the output, the analog-to-digital converters (ADCs) apply time demultiplexing so that the output clock is compatible with the electronic memory. Overall, the speed advancement of the photonic chip can be realized provided the usage of multiplexing DACs and demultiplexing ADCs.”

Comment 6:

Minor concerns include:

The article has quite a few grammar errors that need revision before publication.

Response:

As suggested, we have thoroughly proofread the manuscript and revised grammar errors.

Comment 7:

GBaud/s is the wrong unit. It should read Gbaud.

Response:

Thanks for the error correction. The unit is revised.

Comment 8:

Comment on bit resolution of analog representation. Only a basic simulation done at the output attempted to study the effect of low-precision analog representation, or noise overall, on the recognition accuracy. Can you comment on the control challenges of setting weights with MRRs and its sensitivity to crosstalk and environmental fluctuations?

Response:

As suggested by the reviewer, we add an experimental measurement and comment of the bit resolution of MRR controlling. The major limit for bit resolution is the cross-talk between MRRs. The standard deviation of weight control is 0.035, corresponding to the bit resolution of 4.8 bits at the fixed temperature of 28 °C.

(Lines 28-33, Page 8): “The heat sink inside the packaging module is controlled by a thermal controller (Thorlabs ITC4001), maintaining the chip temperature at 28°C. At this temperature, the standard deviation of MRR weight control is 0.035 out of range 1, corresponding to 4.83 bits of resolution. The resolution is majorly limited by the MRR crosstalk. Without temperature

maintenance, the resonance of MRR will significantly deviate, degrading the bit resolution. Recent approaches to accurate MRR weight controlling [44] are beneficial for the PTFP.”

44. W. Zhang, C. Huang, H. Peng, S. Bilodeau, A. Jha, E. Blow, T. Ferreira de Lima, B. J. Shastri, and P. Prucnal, Silicon microring synapses enable photonic deep learning beyond 9-bit precision, Optica 9, 579-584 (2022).

To Reviewer #2:

Black: Comments from the reviewer.

Blue: Response of the authors.

Red underlined: Revisions to the manuscript.

In the article, "High-order tensor flow processing using integrated photonic circuits," the authors present their work which uses temporal and frequency space to achieve convolutional operations with low latency. Overall the authors have demonstrated an impressive work that uses time delay lines (the main novelty of this manuscript) to enable summation using the time domain. The authors have introduced the topic well and have cited the relevant literature in the introduction. I also have to applaud the authors for the very detailed supplementary information which contains the information needed to replicate their results. I support publication in Nature Communications after the authors address the following concerns:

Response:

Thanks a lot for your comments. We are happy to have the chance to improve the quality of our work following your suggestions. We would like to address your concerns one by one in the following sections.

Comment 1:

The main issue I have with this work is the challenge in understanding how time, frequency, and MRRs all map to the input features and convolutional kernels. The authors have the tensors represented in abstract diagrams in figure 1. It would be very helpful to the reader if the authors could instead represent these tensors as actual matrices like in some of the papers they cite (e.g., similar to Fig 1c in Feldmann et al., Nature 589, 52-58 (2021)). Even now reading the authors' manuscript, I find it very challenging to understand how the time dimension was used. For example, I'm not sure how the authors are implementing the 3x3 kernel shown in Fig. 3g since this cannot be simplified to a single 1x3 kernel used 3 times (like in 3d,e, and f).

Response:

You're appreciated for reminding us of the difficulty of understanding the previous manuscript.

Following your suggestion, we've revised Fig. 1 as well as the description of the principles. The difficulty of understanding the term 'time dimension' is also concerned by Reviewer #1. Therefore, we have revised 'time dimension' to 'delay steps' in the manuscript to clarify definitions. We hope this can address your concern.

The revisions done to the manuscript are enumerated as follows.

(Fig. 1):

Fig. 1 before:

Fig. 1 revised:

(Caption of Fig. 1): “Fig. 1. Basic principles of the PTFP.... In the PTFP approach (marked with ‘Flow’), the input tensor is reshaped and enters the PTFP in serial. Each input channel is temporally modulated onto an individual wavelength. A line in the PTFP schematic represents a convolutional operation between an input channel and an output channel. Inside each line, signals are delayed, weighted, and summed so that a temporal convolution (a.k.a. FIR filter) is completed. An output channel is yielded by combining convolved signals from different input channels...b, Conceptual schematic of the PTFP chip...The directional couplers and delay lines perform data duplication and shifting in the optical domain. Multiple wavelengths are split and delayed in parallel...”

(Lines 28-35, Page 3): “In the process of the PTFP (shown in the ‘Flow’ part), the input tensor is not duplicated. Different input channels are carried by different optical wavelengths. Serial pixels in a single channel are temporally modulated onto the time steps of an optical signal. Inside the PTFP, each input channel is connected with each output channel through a convolutional operation (a line in the figure). A convolutional operation is essentially a finite impulse response (FIR) filter; therefore, we can implement such FIR filters by imposing delaying, weighting, and summation to the input temporal sequence. The number of delay steps is equal to the size of kernel, D_{kernel} .”

(Lines 2-6, Page 4): “Input optical sequences of different wavelengths are firstly combined with a WDM. Then, directional couplers and ODLs are deployed to provide the time delay steps D_t .

In each delay step, optical sequences are further split to provide the dimension of space, D_s . In a specific delay step and space dimension, a weighting bank with $D_w (=C_{in})$ copies MRRs is exploited.”

Regarding the 3x3 kernels of Fig. 3d-g (now Fig. 3f-3i), they are accomplished with three individual 1x3 kernels on three wavelengths. Details are stated in the revised manuscript.

(Lines 18-31, Page 5): “As we have multiple channels for input, we validate the multi-channel convolution in this part... In the case that three input channels are configured with an identical image with row shifting, the multi-channel convolution can equivalently perform a 3×3 convolution...the equivalent 3×3 convolution is a good way to benchmark the multi-channel convolution of this chip so several 3×3 convolutions are demonstrated in Figs. 3e-3i.”

Comment 2:

Since the authors are delaying the optical signal from coherent laser sources, they are forced to use high speed electronics ("EPCs") to recombining the signals electrically to prevent unwanted optical interference. This could be a limitation of this approach as RF power combiners can be bulky and limited in bandwidth (esp. flatness in amplitude and phase). The authors should discuss this and possible routes to address this in on-chip integration.

Response:

This is truly an important issue to discuss. Following your suggestion, we discuss this issue and possible solutions in the revised manuscript.

In our opinion, we have at least two possible routes. The first one is using integrated wide-band EPCs and hybrid photonics-electronics packaging technologies to miniaturize the system (the same idea as packaging TIAs with PDs). From our knowledge, commercially available EPCs support over 40-GHz bandwidth or 80 Gbaud symbol rate, so the adoption of electronics may not be a severe limit on computing speed. The second route is to place PDs compactly and connect the outputs of PDs directly. This approach is demonstrated recently by F. Ashtiani, et al, Nature, 2022, doi:10.1038/s41586-022-04714-0. As long as the PDs are placed close enough (sub-millimeter distance), the photocurrent obeys Kirchhoff's current law so that the outputs are added without an EPC.

Revisions:

(Lines 4-9, Page 9): “Although the adopted EPC is now bulky, there are two routes to enable on-chip miniaturization of the electrical signal combination. One is packaging wideband EPC dies (over 40-GHz bandwidth is commercially available) with the photonic chip using wire bonding or similar technologies. Another is to place the PDs compactly and connect their output wires directly. As long as the PDs are close enough (sub-millimeter distance), the photocurrent obeys Kirchhoff's current law so that the outputs are added without an EPC [32].”

32. F. Ashtiani, A. J. Geers, and F. Aflatouni, An on-chip photonic deep neural network for

image classification, Nature 606, 501-506 (2022).

Comment 3:

The authors mentioned duplication of data and memory access bottlenecks when performing convolutions on other platforms. This can be addressed by better pipelining of weights and input data (for example, the Caffeine optimizer demonstrated in 2016, DOI:10.1109/TCAD.2017.2785257). The authors should discuss more up-to-date methods for implementing CNN in hardware so as not to construct an overly simplistic view of how digital hardware actually performs such operations (for example, full "memory access" for each time data is reused is not the case).

Response:

Thanks for your suggestion. Several state-of-the-art digital methods of implementing CNN are cited and commented on in the revised manuscript.

(Lines 14-20, Page 2): “Given the fact that tensor convolution, especially in the AI field, is consuming an increasing portion of computing resources, high-throughput and energy-efficient processors are desired [14]. Digital methods including generalized matrix multiplication (GeMM) [15], domain transformation [16], and input/weight reusing [17] are investigated to realize high-performance computing (HPC) of tensor convolution. These methods pursue a balanced and optimized performance under limited hardware resources (e.g. memory, bandwidth, and power). Among these methods, GeMM is widely adopted for its high throughput and high flexibility for AI.”

16. T. Abtahi, C. Shea, A. Kulkarni, and T. Mohsenin, Accelerating convolutional neural network with FFT on embedded hardware, IEEE Transactions on Very Large Scale Integration System 26, 1737-1749 (2018).

17. C. Zhang, G. Sun, Z. Fang, P. Zhou, P. Pan, and J. Cong, Caffeine: toward uniformed representation and acceleration for deep convolutional neural networks, IEEE Transactions on Computer-Aided Design of Integrated Circuits And Systems 38, 2072-2085 (2018).

Comment 4:

If I understand correctly, the main use of the time delay is for convolutions with a stride of 1 where the input data is duplicated. If that is the case, convolutions which have a stride equal to the width of the kernel do not have data duplication (like Supplementary Fig. 19). I don't believe this was mentioned or motivated in the main text, so the authors should discuss the specific cases where their time delay method is useful for CNNs.

Response:

You are correct about the functionality of the photonic chip. In the cases where the stride is less than the kernel size, our approach is advantageous since it avoids data duplication compared with conventional GeMM.

Following your suggestion, we've clarified this issue in the manuscript.

(Lines 22-25, Page 3): “In order to compute tensor convolution with the stride of ‘1’, shown by the ‘GeMM’ part of Fig. 1a, GeMM firstly transforms the input tensor to an input matrix with the dimensionality of $[C_{in} \times D_{kernel}, D_{data}]$, where data volume is augmented by D_{kernel} times.”

(Lines 22-23, Page 7): “The PTFP occupies less memory than the mainstream GeMM processors when the stride is smaller than the kernel width.”

To Reviewer #3:

Black: Comments from the reviewer.

Blue: Response of the authors.

Red underlined: Revisions to the manuscript.

The authors of this manuscript claim the demonstration of an integrated photonic tensor processor that performs operations on higher dimensional tensors without the need for conversions to matrices beforehand. This is presented as the key novelty of the paper. They claim to exploit various degrees of freedom in optics such as wavelength, time and space to do so and report on 20 GHz operation which is used in an image sequence classification benchmark, yielding 97.9 % accuracy. This is done with the photonic circuit performing convolution operations as part of a larger CNN implemented with conventional computer hardware. Furthermore, the authors claim comparable performance of the photonic solution to electronic counterparts with the potential to even surpass that.

In my opinion, most of the claims are supported by the provided data and evidence, albeit some claims are formulated in a too strong manner and need to be revised. More detailed comments about that will be given later in this report. The main experimental results seem to be obtained carefully with the methodology outlined in detail both in the main manuscript and in the supplementary material. Various questions are still left open which will be described later in this report and should be addressed by the authors. The experimental data and evidence is presented in a structured and consistent way, the supplementary material is well organized and very complete.

Response:

You are greatly appreciated for your detailed review, which is really helpful for us to improve the quality of our work and manuscript. We would like to address your concerns one by one in the following sections.

Comment 1:

The main issue that needs clarification is the claim of the photonic processor not relying on memory access and tensor to matrix conversions in comparison to other work out there. However, throughout the methodology section, it becomes clear that several data conversion steps (e.g. 2D image to 1D vector conversion among many others) are still needed. In addition, the size of the kernel in experiment is 1 in the first dimension which seems ill-chosen given the major point of the solution to be able to deal with higher-order tensor processing directly in the photonic chip. A 2D kernel operation is described on page 5, 2nd paragraph, and seems to

involve multiple iterations of the hardware. It would have been a much more convincing result if the spatial part of the chip is extended to deal directly with 2D kernels. At least, that would be expected from the way the claim is formulated. Furthermore, memory is in my opinion still used during the operation to set the MRR banks to the desired kernel values.

Response:

Thank you again for reminding us of the problems in our previous manuscript. We would like to respond to your ‘*Comment 1*’ from two aspects (data conversion and kernel dimension).

(1) As you commented, data reshaping is still needed in our architecture. The functionality of the PTFP is reducing memory use of the input data because it removes input data duplication of conventional GeMM. The memory for weights is the same as the conventional approaches. We have revised Fig. 1a to show the exact functionality of our architecture in a more precise way. In addition, according to your suggestion, we have revised the claims in our manuscript to describe our motivation in a more precise way.

In specific, the revisions done to our manuscript are enumerated as follows.

(Fig. 1):

Fig. 1 before:

Fig. 1 revised:

(Lines 1-3, Page 3): “Here, we present an integrated photonic tensor flow processor (PTFP) that processes high-order tensors without digital data duplication and shifting; therefore, no excess memory is occupied for input data preparation.”

(Lines 36-38, Page 3): “In other words, the PTFP approach saves D_{kernel} times of memory for input tensor transformation. The size of memory for weights is the same as conventional GeMM.”

In our mind, the term ‘many others’ in your comment may be the waveform encoding and decoding conversion. It is indeed a compromise that we made under a limited experimental condition. As you have noticed, the insertion loss of the current chip is quite high. We are forced to use electrical amplifiers to increase modulation depth so that the output signal is detectable. In the revised manuscript, we provide several routes and corresponding simulations to circumvent encoding and decoding in future engineering.

(Lines 14-16, Page 9): “However, due to the inferior loss performance of the fabricated chip (see Suppl. note S5 for details), we adopt electrical power amplifiers in our experimental setup to enhance modulation depth so that the output signal is detectable.”

(Lines 10-17, Page 10): “Given that the encoding and decoding processes introduce data transformation overhead and undesirable signal distortion, we pursue removing the encoding and decoding in future works. Using low-drive modulators (such as the lithium niobate film

modulators [37]) will significantly refine the modulation depth, so the electrical amplifiers and encoding/decoding process can be removed. Also, refining the loss performance of the chip benefits the output signal power. Suppl. Fig. S17 shows the effect of insertion loss and modulation depth on the output signal quality. It is found that low-drive modulators and low-loss photonic chips promise good signal quality without data transformation overheads.”

(Suppl. Fig. S17):

(Caption of Suppl. Fig. 17): “Suppl. Fig. 17 Signal quality (SINAD) with different modulation depth and insertion loss. The result is simulated with identical conditions in the experiment by assuming different insertion losses. ‘IL=30 dB’ represents the performance of the current photonic chip. The output signal quality can be refined by both higher modulation depth and better loss performance. In the cases that IL is low, too large modulation depth leads to a signal quality degradation because of the nonlinearity of electrooptic modulation.”

(2) Regarding the 2D convolution, we agree that we should clearly show the experimental limitations and revise the claims that we made in this part. In fact, we conducted an additional experiment to address your concern, which will be described in detail in the response to ‘Comment 4’. Also, we provide the route toward realizing 2D convolution directly on-chip in the revised supplementary notes.

Revisions are made as follows.

(Lines 19-21, Page 4): “The fabricated chip holds the dimensions of $[D_w=4, D_i=3, D_s=1]$. It can conduct multi-channel convolutions with 3 parameters in each channel.”

(Lines 18-31, Page 5): “As we have multiple channels for input, we validate the multi-channel convolution in this part.... Because 3 delay steps are implemented on the chip, these 1×3 convolutions are performed without data duplication. ...In the case that three input channels are configured with an identical image with row shifting, the multi-channel convolution can equivalently perform a 3×3 convolution. We should note that the current PTFP chip is not able to process 3×3 convolution directly without data duplication. However, the equivalent 3×3

convolution is a good way to benchmark the multi-channel convolution of this chip so several 3×3 convolutions are demonstrated in Figs. 3e-3i.”

(Lines 15-30, Page 6 to Lines 1-7, Page 7, Suppl. information): “Currently, the fabricated chip comprises 3 delay steps. By augmenting the number of delay steps, the PTFT can process larger convolutional kernels without data duplication. One way is to cascade more ODLs with the same length. It can enlarge the kernel size of one-dimensional convolution. The other is to introduce long ODLs to form higher-dimensional convolutions. Take two-dimensional image convolution ($\sigma \times \sigma$) for example. If the image is input row by row, the required delays are not uniform. ...The structure to implement such optical delays is shown in Suppl. Fig. 20b. Unit delay shift the neighboring pixels and the long delay shifts the pixels across rows. In the example of 10×10 image, assuming that the clock frequency is 20 GHz, the required delays for different time dimensions are 0 ps, 50 ps, 100 ps, 0.5 ns, 0.55 ns, 0.6 ns, 1 ns, 1.05 ns, 1.1 ns, respectively. Since the delay lines in the PTFP chip is cascaded, the total length of the optical delay lines is determined by the maximal delay. Even if larger kernels and larger images are processed, the required length of optical delay lines is at meter-long level. Given that current technologies of low-loss silicon nitride waveguide support the insertion loss of 1.0 dB/m, long delay of optical signals is technically feasible.”

Significance of the work

The presented work is of high significance as to the best of my knowledge it demonstrates parallel processing of tensors reflecting multiple input channels by a photonic circuit. Although there is a discussion to be had on the dimensions of the chosen data per input channel, the results show that the photonic solution can deal with multiple channels at the same time through wavelength multiplexing, which provides an advantage over matrix multiplication hardware as such. Within each input channel the high speed of the photonic solution is exploited to be able to deal with the information in a serial way.

Data and methodology

Extensive experimental data is provided for the subcircuits of the photonic chip and supported by simulations. Furthermore, data for validating the convolution operation and demonstrating it in a benchmark is given. Points as listed below should be addressed by the authors however:

Comment 2:

The current data to proof the convolution operation is given in Fig. 3c. However, the input data is not provided and it is difficult to judge the correctness of the operation. Indeed, the calculated result is overlaid and it has been made clear that the experimental clock rate leads to a sampling of the ideal data. However, more discussion on the conditions of this sampling is needed. Is the sampling sufficient and how close should the experiment be to calculated result so that it is sufficient? Are we operating at the limit and would a higher sampling rate improve

performance?

Response:

According to your suggestion, we add the input signal in Fig. 3c for reference.

Fig. 3c before:

Fig. 3c revised:

Also, an additional description of sampling is provided in the revised manuscript. The sampling rate is set at 20GS/s, equal to the modulation rate. The sampling is at the Nyquist limit, so the prerequisite for obtaining good results is synchronizing signal generation and sampling accurately (described in section 3 of Methods). A higher sampling rate benefits the accuracy of synchronization but not the performance of results. Because, after an accurate synchronization, the deviation between the experimental result and the calculated one is majorly caused by random noise and signal distortion.

This issue is addressed in the revised manuscript.

(Lines 14-17, Page 5): “From the zoom-in plot, we observe that the experimental results are close to the theoretically calculated samples, verifying the correctness of conducting one-

dimensional convolution. The deviation between the experimental result and the calculated one is mainly caused by experimental noise and waveform distortion.”

(Lines 9-12, Page 8): “The sampling rate of the oscilloscope is configured at 128 GSa/s and the obtained data is down-sampled to 20 GSa/s to reconstruct the results. Note that a high sampling rate allows us to perform input and output synchronization more precisely but a sampling rate of 20 GSa/s is enough to carry out convolutions after synchronization is done.”

Comment 3:

In Fig. 3d-g the input image is not shown, which would give better means to compare the output from the convolution. Why were different input images chosen for this validation? Please explain the reasoning behind.

Response:

According to your suggestion, we add the input image as Fig. 3e for reference. The input image for all convolution examples is the same but Figs. 3h and 3i are the zoomed-in plots for better observation.

Fig. 3 before:

Fig. 3 revised:

Comment 4:

It seems that the results 3d-g are obtained by exploiting different channels to yield a 2D 3x3 kernel. This relates to the main issue mentioned above. Given the claim, one expects different input channels and 2D kernels simultaneously and not used to compensate for each other. Please revise the claim in a way that better reflects the experimental limitations here.

Response:

Thanks a lot for your comment. As we realize that the original claims and experimental result may confuse the readers, we carry out an additional experiment to demonstrate the multi-channel convolution in its supposed way, i.e. conducting three 1D convolutions with three individual input channels. Three different images from a video clip are selected as the input channels. A vertical edge detection kernel $[-1, 0, 1]$ is adopted for each of them. Therefore, the output of the multi-channel convolution should be the superposition of the vertical edges of these images. Additional Fig. 3d shows the result. The subplots on the left are the input channels

and the subplot on the right is the convolved image. Vertical edges of objects are obtained.

Fig. 3d:

Besides, we've revised the claims in this part to better reflect the experimental conditions. (p.s. There are some overlaps with the revisions in "Comment 1")

(Lines 18-31, Page 5): "As we have multiple channels for input, we validate the multi-channel convolution in this part. Three different images from a 'traffic camera' dataset [36] are chosen as the input channels and a vertical edge detection kernel $[-1, 0, 1]$ is adopted for each of them. Therefore, the output of the multi-channel convolution should be the superposition of the vertical edges of these images. Because 3 delay steps are implemented on the chip, these 1×3 convolutions are performed without data duplication. Fig. 3d depicts the result. Three images including a car in each are processed by three wavelength channels and the output shows all vertical edges of these cars. The 'leaves' on the ground are static for three images. The vertical edges of them accumulate three times so that they are very bright in the output. The result confirms the capability of multi-channel convolution. In the case that three input channels are configured with an identical image with row shifting, the multi-channel convolution can equivalently perform a 3×3 convolution. We should note that the current PTFP chip is not able to process 3×3 convolution directly without data duplication. However, the equivalent 3×3 convolution is a good way to benchmark the multi-channel convolution of this chip so several 3×3 convolutions are demonstrated in Figs. 3e-3i."

Comment 5:

In page 6, supplementary material, the authors point to meter long delay lines for larger kernels as those scale with kernel size. It is hinted that those can be implemented using fiber. Given the direction of integration and argumentation of high throughput density, anything off chip would be counter intuitive. Please address the issue of scalability of the kernel and connected with that the limitations to chip area, throughput density, due to delay lines. Also link that with the author's discussion on scaling S7. Furthermore, it is difficult to understand how 'advanced electrooptic modulators, PDs, and electrooptic packaging technologies are beneficial to shortening on-chip ODLs'. What is the relation here?

Response:

Thanks a lot for your comments. As you commented, the adoption of long delay lines inevitably deteriorates throughput density. Therefore, we have revised the discussion about the scalability of the photonic chip by considering the negative impacts of on-chip and off-chip delay lines. An additional table (Suppl. Table 5) is provided for the discussion about ODL area occupation and computing density.

Regarding the off-chip fiber delay, we think it is an optional route for future engineering. The photonic chip is typically installed onto a substrate or a PCB, and the optical input signals are typically fed into the chip with a fiber array. Therefore, the introduction of off-chip fiber delay lines may not obviously increase the total volume of the electro-optic system. Moreover, with the development of low-loss fiber-chip coupling (e.g. photonic wire bonding) technologies, the off-chip fiber delay lines may be less lossy than on-chip delay lines.

This issue is stated more clearly in the manuscript with the following revisions.

(Lines 2-15, Page 8, Suppl. information): “The on-chip delay lines are obviously an obstacle for achieving very high computing density, so we provide the possible routes to diminish the influence of on-chip delay lines. Suppl. Table 5 shows several examples of area occupation of ODLs and the computing density of the chip. The clock rate is set to 20 Gbaud and the width of the input image is set to 64. The Number of input channels is equal to the number of wavelengths. The number of output channels is set to 32, 11, and 6 for 3×3, 5×5, and 7×7 kernels, respectively. Specifically, the area occupied by the Si₃N₄ ODLs is 6.25 mm², 15.08 mm², and 24.55 mm² for 3×3, 5×5, and 7×7 kernels, respectively. It is compatible with off-the-shelf photonic integration. It is seen that more MRRs integrated on the chip will dilute the area occupation of ODLs. Moreover, using the off-chip fiber delay approach described in Suppl. note S6 may be another route. It does not introduce a significant volume increase to the system but lowers the challenge of on-chip delay lines. Among all examples listed in Suppl. Table 5, the largest chip footprint is (24.55+64×7×7×6×0.0144)=295.5 mm², compatible with current photonic integration technologies.”

(Lines 5-12, Page 7, Suppl. information): “Given that current technologies of low-loss silicon nitride waveguide support the insertion loss of 1.0 dB/m, long delay of optical signals is technically feasible. Another possible route for implementing long delays is using off-chip fibers as delay lines. The photonic chip is typically installed onto a substrate or a PCB, and the optical input signals are typically fed into the chip with a fiber array. Therefore, the introduction of off-chip fiber delay lines may not be an obviously increase to the total volume of the system. Moreover, with the development of low-loss fiber-chip coupling (e.g. photonic wire bonding [41]), the off-chip fiber delay lines may be less lossy than on-chip delay lines.”

(Suppl. Table 5):

Kernel size	Area of on-chip delay lines [mm ²]		On-chip computing density [TOPS/mm ²]					
	All on-chip	Fiber-hybrid	All on-chip			Fiber-hybrid		
			16λ	32λ	64λ	16λ	32λ	64λ
3×3	6.25	1.93	2.538	2.652	2.714	2.699	2.737	2.757
5×5	15.08	6.44	2.263	2.494	2.628	2.532	2.649	2.712
7×7	24.55	11.59	2.038	2.351	2.547	2.371	2.558	2.663

Furthermore, the statement on the relationship between modulators and PDs, and the length of optical delay lines is revised for better understandability.

(Lines 35-40, Page 6): “The length of ODLs is inversely proportional to the clock frequency. Advanced electrooptic modulators [37, 38] and PDs [39] with large bandwidth allow higher clock frequency: the length of ODLs is thus shortened... Also, advanced electrooptic packaging technologies [41] offer us the opportunity to use off-chip fibers to replace on-chip long delay lines (discussed in Suppl. note S6).”

Comment 6:

In S7, the throughput is calculated. I don't understand the argument of 2 operations per clock. Given an information coding of 20 GBaud, that means 20E9 values of pixels per second which would also yield 20E9 operations per second on those pixels. I understand the multiplication with the kernel of 12 weights. Please address this.

Response:

For a single clock cycle and a single computing unit, multiplication and addition are conducted. So, the throughput is calculated with a factor of 2. This is a common calculation method adopted in relevant works [such as J. Feldmann, et al, Nature, 2021, doi:10.1038/s41586-020-03070-1]. This issue is addressed in the revised manuscript.

(Lines 23-25, Page 7, Suppl. information): “The PTFP chip conducts multiply and accumulation (MAC) operations, corresponding to 2 operations (a multiplication and an addition) per clock cycle per unit.”

Comment 7:

On page 8 of the manuscript, the input coding is explained that is used to circumvent the bandpass characteristic of the used amplifiers. Given that the sign is changed between each consecutive bits, this means that only one sample exists per symbol. I assume that after the analog modulation, the analog signal stays at those values for the duration of the bit period. How does the sampling affect the processing of the signal. Are we operating at the limit of nyquist sampling here?

Response:

Your understanding is right. The most important issue of sampling is maintaining temporal accuracy because only one position on the analog waveform is the accurate value for one bit. Therefore, the signal generator and the ADC are synchronized in our experiment to realize accurate sampling. The ADC is working at the sampling rate of 128 GS/s and data are down-sampled to 20GS/s. Therefore, the system is working at the limit of Nyquist sampling.

We address this issue in the revised manuscript.

(Lines 9-10, Page 8): “The sampling rate of the oscilloscope is configured at 128 GSa/s and the obtained data is down-sampled to 20 GSa/s to reconstruct the results.”

(Lines 21-23, Page 10): “Moreover, the adopted sampling rate is 20 GSa/s (down-sampled from 128GSa/s), equal to the modulation rate. Only if the sampling clock and the signal generator are synchronized, correct results can be sampled.”

Comment 8:

In same section, the authors write 'Since the error of waveform distortion is relevant to the waveform itself, its influence on the final recognition accuracy may be less than the experimental noise'. This is not clear at all. Are there two aspects that now distort the signal, one being frequencies <10 Ghz not being passed by the amplifier and the second being the coding scheme not having signal components below 0.5 GHz as Fig. S15 shows? How do each of those aspects influence the recognition of the initial images?

Response:

Thanks a lot for the comment. We have revised this part for better understandability of the error and the recognition accuracy in our experiment.

There are two major sources of error in the experiment: one being experimental noise; the second being the waveform distortion because of the amplifiers' low-frequency cutoff. In other words, the error power in the experiment is the combination of noise power and distortion power. Fig. 4f shows that although the standard deviation of combined error is 0.1, the recognition accuracy is a little bit higher than that of pure noise at the standard deviation of 0.1. This indicates that waveform distortion is less impactful than noise on recognition accuracy. Since the above statement is an empirical observation, not a strictly proven conclusion, we clarify that in the revised manuscript.

(Line 1-9, Page 10): “It is obvious that the temporal waveform is distorted: there are many relaxation oscillations compared with the idea result. The reason is that low-frequency component of the waveform is filtered out. It indicates that the error of the convolved results not only comes from the experimental noise, but also the waveform distortion originating from the frequency response. In other words, the error power in the experiment is the combination

of noise power and distortion power. Fig. 4f shows that although the standard deviation of combined error is 0.1, the recognition accuracy is a little bit higher than that of pure noise at the standard deviation of 0.1. This gives an empirical observation that waveform distortion is less impactful than noise on recognition accuracy.”

Comment 9:

Please explain in more detail the reasons to apply cropping and pre-processing to the KTH dataset. It is not clear why that is needed.

Response:

In the KTH dataset, people doing actions are not always at the center of the image. If the video frames are not cropped to place people at the center, the fully-connected layer will perform severe overfitting. Also, there are walking and jogging actions in the dataset. The person can walk out of the image. The pre-processing is to filter out those video frames with no person in them, otherwise, the network will also perform severe overfitting.

To address this issue in the manuscript, revisions are enumerated as follows:

(Lines 38-40, Page 10): “Because the recurrent layer and the fully connected layer are sensitive to image shifting, a small movement of the action subject (i.e. the person) may lead to severe overfitting.”

(Lines 9-11, Page 11): “For moving actions (walking, jogging, and running), if the subject moves out of the image, the CNN will also overfit the datasets. The purpose of preprocessing is to pick up frames with subjects inside the image.”

Comment 10:

On page 10 the authors state that overfitting does not affect the recognition accuracy. I think this is not true. Please revise that part. Data shown in Fig. S17 clearly indicate overfitting. As such it is maybe only a problem for the computer baseline and not relevant to its application on the photonic solution.

Response:

Indeed, the statement ‘overfitting does not affect recognition accuracy’ is not precise. We originally mean that the minor overfitting of training is tolerable since the recognition accuracy is not affected very much. To clarify that, we have revised Suppl. Fig. 19 (previous Suppl. Fig. 17) by adding two more curves of training accuracy and validation accuracy into it.

(Suppl. Fig. 19):

Before:

Revised:

(Lines 14-15, Page 12): “It is observed that the recognition accuracy is not obviously affected by this minor overfitting, so we adopt the trained parameters at 125 epochs as the final parameters for inference.”

Comment 11:

When applying weights to the photonic chip, what was the strategy followed here? Could the authors detail this more? The training will yield a model that is optimized for the digital computer. As the photonic chip is analog, one might expect certain translation might be needed for the weights.

Response:

Sure, we supplement the details of weight loading in the manuscript.

The absolute values of weights are applied with two steps: measuring and recording the

modulation curve (transmission rate vs. applied voltage) of the MRRs; then translating the weights to voltages via the modulation curve. The sign of weights is configured manually with current chip.

(Lines 18-27, Page 8): “The absolute values of weights are applied via two steps: measuring the modulation curves of MRRs (shown in Fig. 2g); translating trained weights to applied voltages. The negative sign is applied equivalently by manually changing the experimental setup. ... We should note that the manual operations for negative weights can be eliminated by using both through and drop ports of the MRRs as demonstrated in [23].”

Comment 12:

Fig. 4f shows the simulated performance in presence of Gaussian noise. Where is the noise added to in specific? Is that to the data signals or weights? Please provide more details here.

Response:

The Gaussian noise is added to the output waveform because we want to reveal the performance of our chip under different noise figures.

(Lines 23-24, Page 6): “In Fig. 4f, we carry out simulations to reveal how the noise in the output signals affects the recognition accuracy (see Methods).”

(Lines 22-23, Page 12): “Since the PTFP chip conducts convolutional layers, we introduce additive Gaussian noise to the output of the convolutional layers.”

Analytical approach

The authors use a simple convolution test to validate the photonic hardware and employ the KTH benchmark to test its real operation. Both methods seem adequate and suitable to be used here. Furthermore, the authors analyse the influence of noise which is important given the analog noisy nature of a photonic solution.

Comment 13:

Improvements

A few aspects as outlined below would improve the understanding of the work.

Could the authors discuss in more detail how many wavelengths can be used in parallel and thus how the amount of input channels can be scaled?

Response:

Sure, the discussion about the feasible number of wavelengths is added.

(Lines 27-29, Page 5, Suppl. information): “The feasible number of wavelengths is dependent on the quality factor of MRRs and the width of the free spectral range. According to [50], the

capable number of wavelengths in a weighting bank can theoretically reach 131.”

Comment 14:

Energy-efficiency is mentioned to be one strong point of a photonic solution. Could the authors include discussion of the energy consumption of the existing solution? So far, no discussion on energy is given and no metrics calculated.

Response:

Energy efficiency is a general pursuit for photonic processor researchers. According to your suggestion, we provide a discussion about the energy efficiency of the photonic solution in the revised manuscript.

The route of achieving good energy performance for our chip is similar to that of other photonic processors: enlarging integration scale, shrinking power use of input (lasers, modulators, and DACs), output (PD amplification and ADCs), and weight tuning.

(Lines 15-21, Page 7, Suppl. information): “Besides the enhancement of functionality, larger integration scale also promises better energy efficiency of photonic processing. The analog photonic processing requires digital/analog interconversions and electrical/photonic interconversions. Compared with digital processors, the usage of DACs, ADCs, lasers, modulators, and photodiodes will consume extra power. As the number of integrated computing unit (MRR in this work) increases, the energy consumed by the interconversions will be diluted. Therefore, the energy efficiency of photonics has the potential to perform superior to its electronic counterparts with large-scale integration [25, 27, 28].”

Comment 15:

The caption in Fig. S8 is wrong and not related to MRRs.

Response:

Thanks for the detailed review. We’ve revised the caption.

Comment 16:

The chip insertion loss is given to be around 30 dB. This is rather high. Could the authors comment on the effect of loss on the actual experiment? In S6, the influence of loss on SNR is discussed but more on a theoretical level. Was the loss a problem in experiment? And how to get to 7.4 dB optical link loss for sufficient SNR as discussed in S6?

Response:

Yes, the current chip is very lossy, resulting in a low SNR at the output waveform. The loss is indeed a problem in our experiment from two aspects. First, we have to use EDFAs before the chip to increase input power, or the output optical power is not detectable. Second, we have to use electrical amplifiers for larger modulation depth to gain better SNR. However, the low-frequency-cutoff electrical amplifiers force us to use encoding and decoding, increasing the complexity of the experiment.

In the future, an insertion loss below 7.4 dB is achievable from various aspects of improvements. For example, use advanced fiber-chip coupling technology (<2dB loss/facet is achievable); use low-loss WDM (<1 dB is achievable); use low-loss delay lines (~1 dB/m is achievable); use on-chip PDs to cancel the output chip-fiber coupling; increase the evenness of the direction coupler array (<1 dB is feasible).

(Lines 15-21, Page 5, Suppl. information): “To compensate for this large loss, we adopted optical amplification (the EDFAs) and electrical amplification (the power amplifier) in the experiment. However, these amplifiers not only increase system complexity and power budget, but also introduce some unnecessary processes such as encoding and decoding. Therefore, in future engineering, the insertion loss of the PTFP chip should be refined significantly from following aspects: using advanced fiber-chip coupling; using low-loss delay lines; using low-loss WDM; increasing the evenness of DCA; using on-chip PD to avoid chip-fiber coupling.”

Comment 17:

Clarity and context

The authors clearly describe the motivation, conceptual framework followed, photonic hardware and experimental results. A few areas could be improved in terms of clarity, as listed below:

The MRR weighting characterization as shown in Fig. 2g is not clear. Why is there a difference between the rings and what is desired?

Response:

As we respond to your ‘comment 11’, the reason for MRR characterization is to record the transmission-voltage mapping to translate weight values to the applied voltages. There are differences for each MRR because of the fabrication deviation. The original resonance point of every MRR varies.

(Lines 33-34, Page 4): “Figure 2g provides the normalized weights (transmission rate) of every MRR with the variation of applied voltages.”

(Caption of Fig. 2): “The original resonance point of MRRs are different because of fabrication deviation.”

Comment 18:

The authors write that the proposed solution allows the signals to 'flow' and perform the needed operation. This concept of flowing is very vague and not clearly defined. Please revise that in the respective sections.

Response:

Thanks a lot for the reminder. We have now described the concept of 'flow' in the introduction. (Lines 3-5, Page 5): "...The serially input data directly enter the PTFP and the output result is yielded serially. Namely, tensor convolution is completed as the input tensor 'flows' through the photonic circuit."

Comment 19:

Fig. 4a depicts the general CNN architecture for the benchmark experiment. Could the authors add the pooling layers and feature maps etc and indicate which parts are performed in photonics and which parts in conventional computer?

Response:

Sure, Fig. 4a is revised according to your suggestion.

(Fig. 4a):

Before:

Revised:

(Lines 3-5, Page 6): "The parameters of CNN are trained firstly on a computer and the PTFP chip is used for computing the 'Conv. 1' and 'Conv. 2' layers in the inference phase (Details of

training are provided in Methods).”

Comment 20:

In the conclusion, the authors state that the photonic solution demonstrates processing with a four-order kernel. Could the authors explain that in more detail to me?

Response:

Thanks for your suggestion. We have revised the statement in the conclusion to precisely describe how the kernel tensor is implemented in the experimental demonstration.

(Lines 23-26, Page 7): “The PTFP chip with multiple wavelengths and delay steps is fabricated for the proof-of-concept experiment, demonstrating a kernel tensor shaped $[1 \times 3, 4, 1]$ for image convolution. With a developed integration scale, the PTFP is capable to implement high-order kernel tensors.”

Expertise

I would like to disclose here the lack of expertise with respect to convolutional neural networks and their in-depth operation to be able to fully assess the methodology and results related to Fig. 4. Furthermore, I am not familiar with the KTH dataset.

REVIEWER COMMENTS

Reviewer #1 (Remarks to the Author):

I have reviewed a previous version of this manuscript (Reviewer 1). My recommendation to the editor was to resubmit an improved version with major revisions to both the evidence and the claims of this paper, and provided detailed commentary on what I felt were the major points. In general, I stand in agreement with the concerns raised by the other reviewers, but I decided to focus on the balance between claims and evidence. In summary, my concerns were three: 1. Insufficient evidence for claim that this processor outperforms electronics by avoiding data duplication and matrix multiplication complexity; 2. Lack of support for high speed (20 GBaud) given that the experiment was not performed in real time; and 3. Insufficient study of the large overhead introduced by A/D and E/O conversion, not to mention a complicated encoding/decoding scheme. The resubmitted manuscript has partially addressed my concerns by either adjusting the claims or clarifying some points in the manuscript. I would like to thank the authors for the detailed response letter, which was easy to follow.

My opinion is that the revised manuscript has made good progress in addressing my concerns, in part by providing justifications due to experimental limitations and delegating solutions to future work. This is an understandable strategy since adapting the experiment and chip design to strengthen the claims would be a lengthy process. That being the case, however, I must insist that the authors revise some of the claims to better reflect the limitations of the current experiment, as reviewer 3 puts it. My revised recommendation to the editor is to request these changes to the manuscript prior to advancing its publication. For example, on page 7, line 15, it can't be said that this work "experimentally demonstrated an integrated photonic tensor flow processor ... capable [of processing] high-order tensor convolutions without extra input data transformation...". Rather, a processor was proposed, and some proof-of-concept demonstration was provided. On page 6, line 31, the PTFP chip cannot be compared in terms of energy density with the NVIDIA A100 GPU because the footprint and energy for the A/D, E/O were not factored in. In my view, this disclaimer must be included to avoid misleading the electronic hardware engineers out there.

More detailed comments following the authors' replies are enumerated as follows:

1.1. OK. I understand the full 3x3 kernel was not implemented experimentally, but the proposal is conceptually clear. It is a shame the main claim of data non-duplication wasn't conceptually demonstrated directly with at least a 2x3 kernel.

1.2 In the "periphery circuit", you require a time multiplexing functionality in the DACs and ADCs. Though true, it is a big challenge if you require signals at 20GBaud. Given that this chip will have a limitation in size (you mentioned elsewhere up to 32 kernels), you need this overhead to be diluted for a fairly low tensor dimension.

1.3 I will accept your argument that the encoding scheme will disappear upon a hypothetical low-drive modulator based on LN on silicon, though that will likely cost you in insertion loss, complexity, and optical nonlinearity.

1.4 You have provided a 7pJ/symbol metric for the aggregate power budget for A/D and E/O conversions, and cited Demirkiran et al., but fell short of including this metric in the main text. I have not found this 7pJ number in the paper you cited and could not evaluate whether it applied to the same A/D, E/O specifications your system requires (20GBaud, "multiplexing" DACs). I recommend you include in the manuscript a reference to the paper you cited, alongside your calculation on the estimated energy per symbol.

2. Thank you for changing the language about the 'time dimension'. I still assert that 'time delay' is not a degree of freedom of light, alongside spatial mode, polarization, or wavelength. (Line 35, page 2)

3. Thank you for clarifying how you performed the negative weight. In my understanding, you

manually configured each experiment twice. Once for non-negative weights, and once for non-positive weights, finally combining results offline. This is obviously not acceptable for a final system, which must run in real-time. Phase-shifting the carrier during encoding is not acceptable either since it does not apply a negative weight – rather, it simply inverts the signal, rendering all effective weights either positive or negative. I think this proposal should be de-emphasized from the manuscript. The balanced PD is a more appropriate solution to the negative weighting problem (as you proposed in prior work) but was not implemented in the current iteration of your chip.

4. Thank you for including this discussion. I think it's most helpful to the reader. I think you forgot to include the total power consumption in the manuscript itself (10W). I would like you to clarify how you arrived at this number. My calculations based on your information was: $20\text{dBm} = 100\text{mW} \rightarrow 100\text{mW} \times 32 \times 3 = 9.6\text{ W}$. Does that mean that your 10W estimate assumes a wall plug efficiency of 100% for the lasers? I think that this is unreasonable and should be significantly increased.

5. Thank you for commenting on the compatibility of memory bandwidth compared to DAC/PIC bandwidth. My view is that your explanation is overly simplistic. In your VR example of multiple 8K/120fps video stream (4GB/s), it would seem to me that it would be impossible to implement in practice, since each video stream (and each decomposed kernel 'recall') would need a different weight configuration. Is the application in which the weights remain constant over multiple frames general enough? Do you require the kernel to be full-sized to reap these benefits? I am imagining a situation where you would dedicate each processor to a convolutional layer in the scheme proposed in Fig. 4. This is a tall order for software developers since weight reconfiguration is much more flexible with GeMM.

8. Thanks for including your precision metric.

A few comments to your responses to other reviewers:

R2C2: Replacing the EPC with connecting PD wires directly is not trivial at the speeds you are considering. An array of M balanced photodetectors would have the capacitance of 2M single ended PD. If your PD is designed to have 20GHz, a 3x3 kernel would likely reduce its bandwidth to 3GHz!

R3C14: You write: "As the number of integrated computing unit (sic) (MRR in this work) increases, the energy consumed by interconversions will be diluted. Therefore, the energy efficiency of photonics has the potential to perform superior (sic) to its electronic counterparts..." The second sentence does not follow the first. Digital processors have no DACs, ADCs, lasers, modulators or photodiodes. You need to demonstrate quantitatively that 1. the power consumed by these components is diluted enough, and 2. the "photonic tensor processor" must also have significantly lower power consumption.

Reviewer #2 (Remarks to the Author):

The authors have provided significant revisions and I recommend publication of the manuscript in its current form. However, one additional tweak to Figure 1 would be helpful. In the current form, it appears that there is no temporal shift of the data illustrated in Figure 1b (labeled "t=0", "t=0", and "t=0"). Shouldn't there be a delay after each photonic delay line (correct labeling should be something along the lines of "t=0", "t=t_delay", and "t=2*t_delay").

Reviewer #3 (Remarks to the Author):

I would like to thank the authors for their effort in addressing my comments and acknowledge that they have thoroughly revised the manuscript, leading to an improved version. The authors give more clarifications to their claims but at times that reduces the significance of their claims and its impact. There are still a few points detailed below that should be addressed by the authors.

(1) The authors have addressed my initial comment 1 by clarifying that three channels are undergoing a 1x3 convolution. They also describe a way to potentially perform a 3x3 kernel convolution but this is not demonstrated with the proof-of-concept device. The reasons are described and understood but again a 3x3 kernel convolution in experiment would be much stronger evidence to support the claim to 'process high-order tensor convolutions'.

To support higher order convolution with the given PTFT, scaling of the ODLs are described as ways to achieve that. However, this seems to be a not scalable solution for integrated optics. In suppl. Fig. 20, the concept for 3x3 kernel is shown. This would require long delay lines that occupy significant space on the PIC. One could argue that the pixel level delay lines can be scaled up to large numbers (albeit calculated to be 14mm on chip already for 100 ps) but it is difficult to achieve that with the row level long delay lines. In addition, the delays need to be accurately timed and no description of any mechanism to do so is given.

(2) The authors revise the text in response to original comment 5 on the scalability of the on-chip ODLs by pointing to off-chip delay lines or using faster modulators and higher throughput which requires shorter delay lines. I agree with the latter. However, I don't think that off-chip delay lines will be able to solve the problem of scalability. They will often be larger in size and more difficult to integrate together with the PIC and PCB.

(3) The authors include a short discussion in response to original comment 14. Yet, no quantification of such is given in the manuscript. Could the authors estimate the energy efficiency of their proposed solution with respect to energy/operation or energy/MAC operation?

(4) In response to original comment 17, the authors add description that the ring resonators for weighting vary due to fabrication tolerances. As they are highly significant for the weighting process, how do you deal with the tolerances in actual operation. Is a calibration/characterization of every MRR needed in advance for such a PTFT solution?

Thank you very much for the constructive comments. Your comments greatly improve the quality of our work and manuscript preparation. The responses to all the concerns are described one by one as follows:

To Reviewer #1:

Black: Comments from the reviewer.

Blue: Response of the authors.

Red underlined: Revisions to the manuscript.

I have reviewed a previous version of this manuscript (Reviewer 1). My recommendation to the editor was to resubmit an improved version with major revisions to both the evidence and the claims of this paper, and provided detailed commentary on what I felt were the major points. In general, I stand in agreement with the concerns raised by the other reviewers, but I decided to focus on the balance between claims and evidence. In summary, my concerns were three: 1. Insufficient evidence for claim that this processor outperforms electronics by avoiding data duplication and matrix multiplication complexity; 2. Lack of support for high speed (20 GBaud) given that the experiment was not performed in real time; and 3. Insufficient study of the large overhead introduced by A/D and E/O conversion, not to mention a complicated encoding/decoding scheme. The resubmitted manuscript has partially addressed my concerns by either adjusting the claims or clarifying some points in the manuscript. I would like to thank the authors for the detailed response letter, which was easy to follow.

Response:

It is greatly appreciated for your efforts on reviewing this manuscript again. Your detailed comments make us realize the problems in the previous manuscript and we are willing to address your concerns one by one in the following sections.

#1

My opinion is that the revised manuscript has made good progress in addressing my concerns, in part by providing justifications due to experimental limitations and delegating solutions to future work. This is an understandable strategy since adapting the experiment and chip design to strengthen the claims would be a lengthy process. That being the case, however, I must insist that the authors revise some of the claims to better reflect the limitations of the current experiment, as reviewer 3 puts it. My revised recommendation to the editor is to request these changes to the manuscript prior to advancing its publication. For example, on page 7, line 15, it can't be said that this work "experimentally demonstrated an integrated photonic tensor flow processor ... capable [of processing] high-order tensor convolutions without extra input data

transformation...". Rather, a processor was proposed, and some proof-of-concept demonstration was provided. On page 6, line 31, the PTFP chip cannot be compared in terms of energy density with the NVIDIA A100 GPU because the footprint and energy for the A/D, E/O were not factored in. In my view, this disclaimer must be included to avoid misleading the electronic hardware engineers out there.

Response:

Thank you very much for your understanding of our experimental limitations. Your sincere advices are quite helpful for improving the quality of paper to match the standard of Nature Communications.

(1) We totally agree with you that we should revise the claims to better reflect the limitation of the current experiment. Now, we have revised the abstract, introduction, results, and conclusion to clarify the proposal of the photonic tensor processor and what is achieved in the experiment. Specifically, the implemented chip and the experiment are described as a small-size proof-of-concept for demonstrating the key innovations of the proposed architecture. A full-size high-order tensor processor are to be implemented based on integration scale development.

(Abstract, Page 1): “Here, we propose an integrated photonic tensor flow processor (PTFP) without digitally duplicating the input data ... In the proof-of-concept experiment, an integrated processor manipulating wavelengths and delay steps is implemented for demonstrating the key functionalities of PTFP.”

(Lines 7-12, Page 3): “In a proof-of-concept experiment, we implement a silicon-based integrated photonic chip to conduct the key functionalities of the PTFP, i.e. the hybrid manipulation of wavelengths and time delays. It demonstrates two-order tensor flow processing and reduces memory use 3 times. Improving the integration scale will upgrade it as a four-order tensor processor and promote memory use reduction.”

(Lines 21-26, Page 4): “The fabricated chip conducts four-channel convolution with 3 parameters in each channel, namely a two-order tensor convolution kernel written as $[D_{kernel}=height \times width=1 \times 3, C_{in}=4, C_{out}=1]$. Given the fact that duplicating the same structure can expand space dimension (C_{out}) and cascading more ODLs will expand ‘height’ and ‘width’ dimensions [33, 34], successful validation of this chip constructs a strong basis for a complete four-order tensor convolution processor.”

(Lines 13-24, Page 7): “In conclusion, we propose an integrated photonic tensor flow processor which is, compared with mainstream GeMM processors, capable to process high-order tensor convolutions without extra input data transformation and memory use. ... In the proof-of-concept experiment, a silicon-based photonic chip is fabricated. It performs a two-order tensor convolution kernel shaped $[1 \times 3, 4, 1]$, demonstrating the key functionalities of the PTFP. Enlarging the integration scale will complete a four-order tensor convolution processor with

currently available photonic integration technologies.”

(2) We entreat the reviewer allowing us to retain the title of the manuscript, since the proposed processor supports high-order tensor processing and the experiment demonstrates the major innovations of the proposal.

(3) We would like to thank you for reminding us to include the disclaimer of how the computing density is calculated in the manuscript. The previous statement may mislead electronic engineers. As you suggested, we delete the comparison with Nvidia A100 and include a disclaimer of computing density calculation.

(Lines 36-38, Page 6): “The computing density of the core part on-chip (electronics excluded) is 588 GOP/s/mm². With a larger scale, the computing density is capable to surpass 1 TOP/s/mm² (discussed in Suppl. note S7).”

#2

More detailed comments following the authors’ replies are enumerated as follows:

1.1. OK. I understand the full 3x3 kernel was not implemented experimentally, but the proposal is conceptually clear. It is a shame the main claim of data non-duplication wasn't conceptually demonstrated directly with at least a 2x3 kernel.

Response:

We really appreciate your understanding of our best try under the current experimental conditions. And we accept your criticism sincerely. Our hope for the full-size four-order tensor processor is the same as yours. We believe the small-size demonstration of data non-duplication concept will motivate interested researchers and engineers. Also, we are still working on the concept and will try our best to complete the four-order tensor processor in a near future.

(Lines 3-8, Page 7): “Based on the validation of the key functionalities of the PTFP concept, it is feasible to improve the performance of integrated photonic devices and the completeness of high-order tensor convolution. Refining the insertion loss of the chip will contribute to a better signal-to-noise ratio. Increasing the integration scale with extra ODLs and weighting banks will upgrade the current chip to a complete four-order tensor processor (discussed in Suppl. note S5 and Suppl. note S6).”

#3

1.2 In the "periphery circuit", you require a time multiplexing functionality in the DACs and ADCs. Though true, it is a big challenge if you require signals at 20GBaud. Given that this chip will have a limitation in size (you mentioned elsewhere up to 32 kernels), you need this overhead to be diluted for a fairly low tensor dimension.

Response:

Thanks very much for giving us the chance of clarifying the overhead of A/D interconversions as the description in the previous manuscript is not clear enough.

We think this concern is relevant to your ‘1.4’ comment. The overhead of A/D interconversion per symbol is calculated as 6.08 pJ (see response #5). For each input symbol, $2 \times D_{kernel} \times C_{out}$ operations are conducted. The number goes larger with larger integration scale, of course, before reaching the upper limitation. In our previous theoretical work [ref. 33], we have envisioned $D_{kernel} \times C_{out} = 32 \times 9 = 288$. Therefore, the A/D interconversion overhead diluted down to $6.08/288/2 = 0.011$ pJ/operation is feasible.

To address your concern, we describe the overhead issue of high-speed A/D interconversions and its dilution with quantitative evaluations.

(Lines 1-3, Page 9): “With the detailed calculation in Suppl. note S7, we show that the high-speed DACs and ADCs will introduce an energy overhead around 6.08 pJ/symbol. With the expansion of integration scale, it is promising to dilute the energy overhead of DACs and ADCs down to 0.011 pJ per operation.”

(Lines 16-29, Page 7 in Suppl.): “Specifically, the FoM of DACs is formulated as $2^{bit} \times fs/P$. A 28nm-CMOS process fabricated DAC has the specifications of 177 mW, 10 GS/s, and 14 bits [S6]. Then, a DAC working at 20 GS/s and 8 bits is evaluated to consume the power of $177/2^{14} \times 2 = 5.53$ mW. For ADCs, the Schreier’s FoM [S7] reads $FoM_S = SINAD + 10 \times \log_{10}(fs/P/2)$. Take an ADC (5 GS/s, 8 bits, 29 mW) fabricated with 28-nm CMOS process as the reference. An ADC working at 20 GS/s, 8 bits may consume power of 116 mW. The A/D interconversion of a symbol introduces $(116 + 5.53)/20 = 6.08$ pJ overhead. ... Assume a $[3 \times 3, 32, 32]$ kernel is implemented on-chip (technically feasible), the energy overhead for a single operation can be calculated as

$$E = \left(\frac{(116 + 5.53) \text{ [mW]}}{20 \text{ Gbaud} \times 32 \times 9} + \frac{7960 \text{ [mW]}}{20 \text{ Gbaud} \times 32 \times 32 \times 9} + \frac{0.317 \text{ [pJ]}}{32 \times 9} \right) / 2 = 0.0327 \text{ pJ.}$$

where the first term is from the A/D interconversion; the second term is from the lasers; the third term is from the E/O interconversions.”

S6. C. Demirkiran, F. Eris, G. Wang, J. Elmhurst, N. Moore, N. C. Harris, A. Basumallik, V. J. Reddi, A. Joshi, D. Bunandar, An electro-photonic system for accelerating deep neural networks, preprint at arXiv: arXiv:2109.01126v1 (2021).

S7. B. Murmann, The race for the extra decibel: a brief review of current ADC trajectory, IEEE Solid-State Circuit Magazine 7, 58-66 (2015).

#4

1.3 I will accept your argument that the encoding scheme will disappear upon a hypothetical low-drive modulator based on LN on silicon, though that will likely cost you in insertion loss,

complexity, and optical nonlinearity.

Response:

We are really happy to see your understanding. As you commented, integration of heterogeneous materials (LNOI and silicon-based materials) is quite challenging, likely increasing IL, complexity and introducing optical nonlinearity to the device. We agree with that and we also have an optimistic view of LNOI modulators. Recently, LNOI modulators are developing very fast. To our knowledge, the LNOI modulators are low-drive, low-loss, wideband, and better linearity over silicon-based modulators. With the breakthrough of co-integration of LNOI with passive silicon photonic circuits (Si/Si₃N₄/SiO₂), significant performance improvement of the PTFP is expectable.

We address this issue in the revised manuscript.

(Lines 21-24, Page 10): “Although integrating LNOI modulators with silicon photonic circuits is challenging and may introduce extra loss, complexity, and optical nonlinearity, recent breakthroughs in LNOI modulators and the heterogeneous integration technologies are quite inspiring for the performance refinement of the PTFP concept.”

#5

1.4 You have provided a 7pJ/symbol metric for the aggregate power budget for A/D and E/O conversions, and cited Demirkiran et al., but fell short of including this metric in the main text. I have not found this 7pJ number in the paper you cited and could not evaluate whether it applied to the same A/D, E/O specifications your system requires (20GBaud, "multiplexing" DACs). I recommend you include in the manuscript a reference to the paper you cited, alongside your calculation on the estimated energy per symbol.

Reponse:

Thank you mentioning the A/D and D/A power budget issue. We agree with you to explicitly show how is the power metric is calculated.

In Demirkiran's, a commonly used calculation method of DAC power budget is provided, i.e. fix the FoM of DACs and then evaluate equivalent power budget with other configurations. An example is given. The FoM of DACs is formulated as $2^{bit} \cdot f_s / P$. A 28nm-CMOS process fabricated DAC has the specifications of 177 mW, 10 GS/s, 14 bits. That is to say, a DAC working at 20 GS/s and 8 bits may consume the power of $177 / 2^{(14-8)} \times 2 = 5.53$ mW. This evaluation method is also general for ADC characterization (please see B. Murmann, IEEE Solid-State Circuits Magazine, 2015, doi: 10.1109/MSSC.2015.2442393). The Schreier's FoM of ADCs reads $FoM_S = SINAD + 10 \cdot \log_{10}(f_s / P/2)$. Take an ADC (5 GS/s, 8 bits, 29 mW) fabricated with 28-nm CMOS process as the reference. An ADC working at 20 GS/s, 8 bits may

consume power of 116 mW. Overall, the aggregate power consumed by a symbol of A/D interconversion is estimated around $(5.53+116)/20= 6.08$ pJ. The power budget of E/O interconversion is given as $20+297$ fJ/bit in C. Sun, et al, Nature 528, 2015, <https://doi.org/10.1038/nature16454>. These values are usable for roughly evaluating PTFP by changing digital modulation to analog modulation.

To address this issue, the manuscript is revised as follows.

(Lines 16-29, Page 7 in Suppl.): “Specifically, the FoM of DACs is formulated as $2^{bit*fs/P}$. A 28nm-CMOS process fabricated DAC has the specifications of 177 mW, 10 GS/s, and 14 bits [S6]. Then, a DAC working at 20 GS/s and 8 bits is evaluated to consume the power of $177/2^{(14-8)*2}=5.53$ mW. For ADCs, the Schreier’s FoM [S7] reads $FoM_S=SINAD+10*\log_{10}(fs/P/2)$. Take an ADC (5 GS/s, 8 bits, 29 mW) fabricated with 28-nm CMOS process as the reference. An ADC working at 20 GS/s, 8 bits may consume power of 116 mW. The A/D interconversion of a symbol introduces $(116+5.53)/20=6.08$ pJ overhead. ...For modulators and PDs, the power budget of a symbol is evaluated at around 317 fJ [S8]. Assume a $[3\times 3, 32, 32]$ kernel is implemented on-chip (technically feasible), the energy overhead for a single operation can be calculated as

$$E = \left(\frac{(116 + 5.53) \text{ [mW]}}{20 \text{ Gbaud} \times 32 \times 9} + \frac{7960 \text{ [mW]}}{20 \text{ Gbaud} \times 32 \times 32 \times 9} + \frac{0.317 \text{ [pJ]}}{32 * 9} \right) / 2 = 0.0327 \text{ pJ.}$$

where the first term is from the A/D interconversion; the second term is from the lasers; the third term is from the E/O interconversions.”

S8. C. Sun, M. Wade, Y. Lee, J. Orcutt, L. Alloatti, M. Georgas, A. Waterman, J. Shainline, R. Avizienis, S. Lin, B. Moss, R. Kumar, F. Pavanello, A. Atabaki, H. Cook, A. J. Ou, J. Leu, Y. Hsin Chen, K. Asanović, R. J. Ram, M. Popovic, and V. Stojanović, Single-chip microprocessor that communicates directly using light, Nature 528, 534-538 (2015).

#6

2. Thank you for changing the language about the 'time dimension'. I still assert that 'time delay' is not a degree of freedom of light, alongside spatial mode, polarization, or wavelength. (Line 35, page 2)

Response:

Thanks again for your language correction. The Hamerly’s paper (ref.31) presents an approach of manipulating time dimension of light pulses, instead of time delays. It’s our mistake and this sentence in introduction is corrected according to your advice.

(Lines 35-36, Page 2): “For example, wavelengths [23, 26, 29], guiding modes [30], time [31], and space [22, 24, 32] are successfully investigated to carry out linear transformations.”

#7

3. Thank you for clarifying how you performed the negative weight. In my understanding, you manually configured each experiment twice. Once for non-negative weights, and once for non-positive weights, finally combining results offline. This is obviously not acceptable for a final system, which must run in real-time. Phase-shifting the carrier during encoding is not acceptable either since it does not apply a negative weight – rather, it simply inverts the signal, rendering all effective weights either positive or negative. I think this proposal should be de-emphasized from the manuscript. The balanced PD is a more appropriate solution to the negative weighting problem (as you proposed in prior work) but was not implemented in the current iteration of your chip.

Response:

You understanding is correct and thank you very much for your advice. In the revised manuscript, the phase-shifting method is de-emphasized. And the approach of using balanced PDs is envisioned and cited.

(Lines 17-24, Page 8): “In the experiment, the negative sign is applied manually by reconfiguring the experimental setup. For the multi-channel image convolution experiment, the negative sign is applied by connecting the optical signal to the negative input port of the balanced PD (Lab Buddy DSC-R412). ... We note that, in the final version of the PTFP system, the manual operations for negative weights should be eliminated by using both through and drop ports of the MRRs with balanced PDs, as demonstrated in [23, 33].”

#8

4. Thank you for including this discussion. I think it's most helpful to the reader. I think you forgot to include the total power consumption in the manuscript itself (10W). I would like you to clarify how you arrived at this number. My calculations based on your information was: $20\text{dBm} = 100\text{mW} \rightarrow 100\text{mW} \times 32 \times 3 = 9.6\text{ W}$. Does that mean that your 10W estimate assumes a wall plug efficiency of 100% for the lasers? I think that this is unreasonable and should be significantly increased.

Response:

We apologize for the unclear explanation of power budget in the previous manuscript. And thank you very much for giving us the opportunity to offer more detailed information in order to avoid misunderstanding.

Firstly, the number of lasers equals the number of input channels (i.e. C_{in}). It is limited by how many wavelengths we can use in an FSR of the MRRs. The number ‘32’ in the previous manuscript denotes the maximum number of output channels (C_{out}) with a specific optical link. It is limited by the affordable optical power after the WDM (20 dBm evaluated) and link

insertion loss because we need enough power in each output channel to achieve enough SNR.

Secondly, the optical power after the WDM (20 dBm) is the aggregation of all laser power, so each laser provides $100\text{mW}/C_{in}$ optical power. In our theoretical work, we use a conservative value of wall plug efficiency, 5%. Also, the insertion loss of modulators is considered as -6 dB (0.251 linearly). The power budget for lasers is then calculated by $(100\text{mW}/C_{in}/0.05/0.251) \times C_{in} = 7.96\text{W}$.

Finally, including other parts of A/D and E/O interconversions (value calculated in response #5), the total power overhead is estimated at 3.88 (A/D) + 7.96 (Lasers) + 0.2 (E/O) = 12.04 W, corresponding to 0.0327 pJ per operation.

This issue is addressed with the following revision.

(Lines 22-24, Page 7 in Suppl.): “For lasers, the wall-plug efficiency is set to 5%. The laser power to support a $C_{out}=32$, $D_{kernel}=9$ convolutional kernel is evaluated as 7.96 W in total. Ref. [33] presents the details.”

#9

5. Thank you for commenting on the compatibility of memory bandwidth compared to DAC/PIC bandwidth. My view is that your explanation is overly simplistic. In your VR example of multiple 8K/120fps video stream (4GB/s), it would seem to me that it would be impossible to implement in practice, since each video stream (and each decomposed kernel 'recall') would need a different weight configuration. Is the application in which the weights remain constant over multiple frames general enough? Do you require the kernel to be full-sized to reap these benefits? I am imagining a situation where you would dedicate each processor to a convolutional layer in the scheme proposed in Fig. 4. This is a tall order for software developers since weight reconfiguration is much more flexible with GeMM.

Response:

Thank you for asking us to elaborate on the parallel video processing issue. We are happy to answer your questions.

If we want to apply an identical algorithm (e.g. facial recognition, action recognition, and object detection) to multiple streams of video, multiple streams of video are usually grouped as a batch. The algorithm and weights for a batch is constant. Therefore, we think this kind of parallel processing is general in practice.

We think the kernel is not necessary to be full-sized but, as you mentioned, larger size will offer better throughput and efficiency performance. If a kernel of CNN layer is too large, it should be decomposed and the PTFP is reconfigured to run multiple times until finishing. This is also the strategy of other processors whether photonics or electronics.

On the issue of reconfiguration flexibility, we totally agree with you that electronic GeMM is advantageous. The PTFP obeys the same rule of other photonic processors (described in the

‘Research opportunity’ part of M. A. Al-Qadasi, et al, APL Photonics 7, 2022, doi: 10.1063/5.0070992): the less reconfiguration, the higher efficiency. So, photonic processors are encouraged to be large-scale integrated and to process large batches within a round of reconfiguration.

These questions are addressed in the revised manuscript.

(Lines 27-31, Page 7): “Enabled by photonic ultrafast clock frequency, the proposed PTFP is advantageous in parallel processing with large-batch data and is promising to promote advances in compute-intense applications such as multi-stream video processing, high-resolution surveillance, autonomous driving, and the Internet of Things.”

(Lines 1-5, Page 8 in Suppl.): “Moreover, the PTFP follows the same rule as other photonic processors do. The flexibility of reconfiguration is lower than digital electronics and each round of reconfiguration consumes extra tuning energy [S9]. The way of reaching high efficiency is increasing the integration scale and enlarging the data batch size so that a large amount of computation is conducted within a round of reconfiguration.”

S9. M. A. Al-Qadasi, L. Chrostowski, B. J. Shastri, and S. Shekhar, Scaling up silicon photonic-based accelerators: Challenges and opportunities, APL Photonics 7, 020902 (2022).

#10

8. Thanks for including your precision metric.

Response:

It’s our luck to have your advices for improving the quality of our manuscript.

#11

A few comments to your responses to other reviewers:

R2C2: Replacing the EPC with connecting PD wires directly is not trivial at the speeds you are considering. An array of M balanced photodetectors would have the capacitance of 2M single ended PD. If your PD is designed to have 20GHz, a 3x3 kernel would likely reduce its bandwidth to 3GHz!

Response:

We sincerely thank your pointing out the key issue of directly connecting PDs. The capacitance will accumulate and degrades bandwidth. So, this approach is only feasible with ultrafast PDs.

This issue is addresses in the revised manuscript.

(Lines 10-12, Page 9): “Note that the direct PD connection will accumulate their parasite capacitance and lower the final bandwidth. Therefore, this approach is feasible with high-speed PDs whose bandwidth is multiple times higher than required.”

#12

R3C14: You write: "As the number of integrated computing unit (sic) (MRR in this work) increases, the energy consumed by interconversions will be diluted. Therefore, the energy efficiency of photonics has the potential to perform superior (sic) to its electronic counterparts..." The second sentence does not follow the first. Digital processors have no DACs, ADCs, lasers, modulators or photodiodes. You need to demonstrate quantitatively that 1. the power consumed by these components is diluted enough, and 2. the "photonic tensor processor" must also have significantly lower power consumption.

Response:

Thanks again for pointing out the energy consumption issue. Following your suggestion, more quantitative metrics are provided instead of simply claiming the 'superior' performance. Detailed metrics are provided in responses #5 and #8.

To Reviewer #2:

Black: Comments from the reviewer.

Blue: Response of the authors.

Red underlined>: Revisions to the manuscript.

#1

The authors have provided significant revisions and I recommend publication of the manuscript in its current form. However, one additional tweak to Figure 1 would be helpful. In the current form, it appears that there is no temporal shift of the data illustrated in Figure 1b (labeled "t=0", "t=0", and "t=0"). Shouldn't there be a delay after each photonic delay line (correct labeling should be something along the lines of "t=0", "t=t_delay", and "t=2*t_delay").

Response:

You are greatly appreciated for the approval of our manuscript. Following your suggestion, Fig. 1b is modified.

(Fig. 1):

To Reviewer #3:

Black: Comments from the reviewer.

Blue: Response of the authors.

Red underlined: Revisions to the manuscript.

I would like to thank the authors for their effort in addressing my comments and acknowledge that they have thoroughly revised the manuscript, leading to an improved version. The authors give more clarifications to their claims but at times that reduces the significance of their claims and its impact. There are still a few points detailed below that should be addressed by the authors.

Response:

Thank you again for giving us the opportunity to improve the manuscript. We sincerely accept your criticism and we will address your concerns one by one in the following sections.

#1

The authors have addressed my initial comment 1 by clarifying that three channels are undergoing a 1x3 convolution. They also describe a way to potentially perform a 3x3 kernel convolution but this is not demonstrated with the proof-of-concept device. The reasons are described and understood but again a 3x3 kernel convolution in experiment would be much stronger evidence to support the claim to 'process high-order tensor convolutions'.

To support higher order convolution with the given PTFT, scaling of the ODLs are described as ways to achieve that. However, this seems to be a not scalable solution for integrated optics. In suppl. Fig. 20, the concept for 3x3 kernel is shown. This would require long delay lines that occupy significant space on the PIC. One could argue that the pixel level delay lines can be scaled up to large numbers (albeit calculated to be 14mm on chip already for 100 ps) but it is difficult to achieve that with the row level long delay lines. In addition, the delays need to be accurately timed and no description of any mechanism to do so is given.

Response:

(1) We totally agree with you that a 3x3 kernel demonstration would be stronger evidence of high-order tensor processing. And we sincerely appreciate your understanding of our best try using current proof-of-concept device. To address your concern about the claims and the experimental demonstration, the abstract, introduction, results and conclusion of the manuscript are further revised. Specifically, the implemented chip and the experiment are described as a

small-size proof-of-concept for demonstrating the key innovations (multi-channel input and data non-duplication) of the proposed architecture.

We wish these justified descriptions on what is proposed and what is experimentally achieved can address your concern.

(Abstract, Page 1): “Here, we propose an integrated photonic tensor flow processor (PTFP) without digitally duplicating the input data...In the proof-of-concept experiment, an integrated processor manipulating wavelengths and delay steps is implemented for demonstrating the key functionalities of PTFP.”

(Lines 7-12, Page 3): “In a proof-of-concept experiment, we implement a silicon-based integrated photonic chip to conduct the key functionalities of the PTFP, i.e. the hybrid manipulation of wavelengths and time delays. It demonstrates two-order tensor flow processing and reduces memory use 3 times. Improving the integration scale will upgrade it as a four-order tensor processor and promote memory use reduction.”

(Lines 21-26, Page 4): “The fabricated chip conducts four-channel convolution with 3 parameters in each channel, namely a two-order tensor convolution kernel written as $[D_{kernel}=height \times width=1 \times 3, C_{in}=4, C_{out}=1]$. Given the fact that duplicating the same structure can expand space dimension (C_{out}) and cascading more ODLs will expand ‘height’ and ‘width’ dimensions [33, 34], successful validation of this chip constructs a strong basis for a complete four-order tensor convolution processor.”

(Lines 3-8, Page 7): “Based on the validation of the key functionalities of the PTFP concept, it is feasible to improve the performance of integrated photonic devices and the completeness of high-order tensor convolution. Refining the insertion loss of the chip will contribute to better signal-to-noise ratio. Increasing integration scale with extra ODLs and weighting banks will upgrade the current chip to a complete four-order tensor processor (discussed in Suppl. note S5 and Suppl. note S6).”

(Lines 13-24, Page 7): “In conclusion, we propose an integrated photonic tensor flow processor which is, compared with mainstream GeMM processors, capable to process high-order tensor convolutions without extra input data transformation and memory use. ...In the proof-of-concept experiment, a silicon-based photonic chip is fabricated. It performs a two-order tensor convolution kernel shaped $[1 \times 3, 4, 1]$, demonstrating the key functionalities of the PTFP. Enlarging the integration scale will complete a four-order tensor convolution processor with currently available photonic integration technologies.”

(2) Your concern about the challenge of realizing row-level long delays are quite important. To address your concern, we would like to elaborate the discussion of implementing long delays on-chip using currently available technologies.

To our knowledge, Si_3N_4 is a promising platform for ultra-low loss waveguides. We simulate and design a row-level (64 pixels) ODL on silicon nitride platform (see below, also attached as Suppl. Fig. S21c). Referred to X. Ji, et al, Optica 4, 2017

<https://doi.org/10.1364/OPTICA.4.000619>, the bending radius is set to 100 microns. Its area occupation is $2.5 \times 2.66 = 6.65 \text{ mm}^2$. Based on this value, the area occupied by ODLs for a 3×3 kernel is $2 \times 6.65 + 6 \times 0.32$ (pixel-level ODLs) $= 15.2 \text{ mm}^2$. For 5×5 and 7×7 kernels, it is 33 mm^2 and 53.4 mm^2 , respectively. Given the die size of the AMF global foundry is $23.1 \times 30 \text{ mm}^2$. We think integration of row-level ODLs is realizable with current technologies. We should note that the number of ODL is irrelevant to the number of input/output channels. For example, a four-order tensor convolution kernel shaped $[3 \times 3, 32, 32]$ also require ODLs occupying 15.2 mm^2 .

As you commented, the accurate timing of ODLs is also an important issue that should be addressed in the manuscript. In practice, the inaccurate design of the group refractive index will cause minor error of actual delay length. As the length of ODL becomes large, the length error may lead to computational error. Typical compensation method for minor length errors is introducing tuning components such as micro heaters (see X. Wang, et al, *Optica*, 507, 2017, doi: 10.1364/OPTICA.4.000507 and X. Ji, et al, *AIP Photonics*, 2019, doi: 10.1063/1.5111164). Large errors should be optimized by changing designs and iterative trials.

To address the above issues, manuscript is revised as follows.

(Lines 18-25, Page 8 in Suppl.): “To realize a two-dimensional convolutional kernel, row-level long ODLs should be integrated on-chip. Based on the silicon nitride platform, we design a long ODL covering 64-pixel delaying. The bending radius is set to 100 microns [S10]. The layout is shown in Suppl. Fig. 21c. It occupies an area of $2.5 \times 2.66 = 6.65 \text{ mm}^2$ The width (L) of the input image is set to 64. So, the area occupied by the Si_3N_4 ODLs is 15.2 mm^2 , 33.0 mm^2 , and 53.4 mm^2 for 3×3 , 5×5 , and 7×7 kernels, respectively.”

(Suppl. Fig. S21c):

(Lines 4-9, Page 7 in Suppl.): “The precision of delay length is impactful to the accuracy of computing. Due to the fabrication error and the deviation of group index during design, the row-level long ODLs may suffer from length imprecision. A typical solution is introducing tuning components such as microheaters [S4, S5] to perform calibration for minor errors. Large errors should be optimized by changing designs and iterative trials.”

S4. X. Wang, L. Zhou, R. Li, J. Xie, L. Lu, K. Wu, and J. Chen, Continuously tunable ultra-thin silicon waveguide optical delay line, *Optica* 4, 507-515 (2017).

S5. X. Ji, X. Yao, Y. Gan, A. Mohanty, M. A. Tadayon, C. P. Hendon, and M. Lipson, On-chip

tunable photonic delay line, APL Photonics 4, 090803 (2019).

S10.X. Ji, F. A. S. Barbosa, S. P. Roberts, A. Dutt, J. Cardenas, Y. Okawachi, A. Bryant, A. L. Gaeta, and M. Lipson, Ultra-low-loss on-chip resonators with sub-milliwatt parametric oscillation threshold, Optica 4, 619-624 (2017).

#2

The authors revise the text in response to original comment 5 on the scalability of the on-chip ODLs by pointing to off-chip delay lines or using faster modulators and higher throughput which requires shorter delay lines. I agree with the latter. However, I don't think that off-chip delay lines will be able to solve the problem of scalability. They will often be larger in size and more difficult to integrate together with the PIC and PCB.

Response:

Thanks a lot for reminding us the problem of using off-chip ODLs. You are right. The preferable way of implementing ODLs should be on-chip, since compactness is a critical specification for processors. The off-chip ODL is bulky and counter-intuitive with integrated processors. In the revised manuscript, we delete all descriptions about the off-chip delay lines approach.

#3

The authors include a short discussion in response to original comment 14. Yet, no quantification of such is given in the manuscript. Could the authors estimate the energy efficiency of their proposed solution with respect to energy/operation or energy/MAC operation?

Response:

We appreciate your advice about the energy consumption, which is also concerned by Reviewer #1. We agree with you that it's better to include a quantitative evaluation of energy overhead in the manuscript.

(Lines 16-29, Page 7 in Suppl.): “Specifically, the FoM of DACs is formulated as \$2^{bit} * fs/P\$. A 28nm-CMOS process fabricated DAC has the specifications of 177 mW, 10 GS/s, and 14 bits [S6]. Then, a DAC working at 20 GS/s and 8 bits is evaluated to consume the power of \$177/2^{(14-8)} * 2 = 5.53\$ mW. For ADCs, the Schreier's FoM [S7] reads \$FoM_S = SINAD + 10 * \log_{10}(fs/P/2)\$. Take an ADC (5 GS/s, 8 bits, 29 mW) fabricated with 28-nm CMOS process as the reference. An ADC working at 20 GS/s, 8 bits may consume power of 116 mW. The A/D interconversion of a symbol introduces \$(116+5.53)/20 = 6.08\$ pJ overhead. For lasers, the wall-plug efficiency is set to 5%. The laser power to support a \$C_{out}=32, D_{kernel}=9\$ convolutional kernel is evaluated as 7.96 W in total. Ref. [33] presents the details. For modulators and PDs, the power budget of a symbol is evaluated at around 317 fJ [S8]. Assume a \$[3 \times 3, 32, 32]\$ kernel is implemented on-chip

(technically feasible), the energy overhead for a single operation can be calculated as

$$E = \left(\frac{(116 + 5.53) \text{ [mW]}}{20 \text{ Gbaud} \times 32 \times 9} + \frac{7960 \text{ [mW]}}{20 \text{ Gbaud} \times 32 \times 32 \times 9} + \frac{0.317 \text{ [pJ]}}{32 \times 9} \right) / 2 = 0.0327 \text{ pJ.}$$

where the first term is from the A/D interconversion; the second term is from the lasers; the third term is from the E/O interconversions.”

#4

In response to original comment 17, the authors add description that the ring resonators for weighting vary due to fabrication tolerances. As they are highly significant for the weighting process, how do you deal with the tolerances in actual operation. Is a calibration/characterization of every MRR needed in advance for such a PTFT solution?

Response:

The answer is yes. Before we adjust weights of the MRRs, their transmission-voltage curves should be measured. Then we can build a look-up table that maps desired weights to applied voltages. This measurement is required once under a static temperature. Using the built look-up table, we can reconfigure the MRRs with any weight combinations.

(Lines 38-39, Page 4): “This weight-voltage mapping is measured once under static temperature and is used for translating kernel weights to the applied voltages.”

REVIEWERS' COMMENTS

Reviewer #1 (Remarks to the Author):

I have reviewed a previous version of this manuscript (Reviewer 1). My recommendation was to request further modifications in the manuscript in the intent of balancing claims and evidence. The authors made significant changes to the manuscript and addressed most of my concerns satisfactorily. I would like to thank the authors for the detailed response letter, once again.

As I expected, though the addressed comments improved the quality of the manuscript, it reduced the overall impact of the paper, because the 1. proof-of-concept experiment does not outperform electronics (though the proposed architecture might), 2. the proposed speed was technically achieved (20GBaud) but since the operation was not done in real-time, this number is misleading relative to digital processors; and 3. the overhead introduced by the encoding/decoding scheme can become prohibitive to a practical implementation that outcompetes electronics.

With that said, thanks to the modifications in the manuscript, the careful reader will adequately understand the limitations of the PTFP. If the editor is comfortable with this changing scope, I will recommend publication after addressing minor issues below:

1. In your calculation for the ADC/DAC power consumption (Response #5), you used a FOM formula (from Murmann 2015) to estimate how power scales based on a demonstrated device. I'd like to point out that the formula is not directly applicable to speeds above 100MHz (see Fig. 3 in Murmann 2015). Power scales superlinearly with frequency beyond that point. So your calculations are off.
2. I'd like more justification on why the laser power can be scaled with C_{in} . It looks like in your analysis you fix total laser power to 20dBm, so if C_{in} is scaled up to 32 taps, will that provide sufficient optical power to reach appropriate SNR at the photodetector?

Reviewer #3 (Remarks to the Author):

I have reviewed previous versions of this manuscript, pointing out several aspects that need to be revised. The major issues were in the claims made by the authors and the experimental evidence provided, related to the size of the kernel during the PTFP operation. The main novelty was claimed to be the ability of the proposed PTFP to perform higher dimensional tensor operation in hardware utilizing the properties of the photonics chip (wavelength, space, time multiplexing). However, the experiment was performed with a reduced set of dimensions. Furthermore, various other critical points were raised in previous review reports.

The authors have thoroughly addressed the raised issues. The current manuscript has adapted the claims and added explicit explanations on the reduced dimensionality of the PTFP operation in experiment. The authors clarify that the experiment acts as a proof-of-concept and reduce their claim to the proposition of such a PTFP concept. The authors have added additional information on power consumption, delay line scaling, data processing among others which raises the quality of the current manuscript.

With the reduced claim, the impact of the manuscript is also reduced. However, the main novelty to exploit different wavelengths for the processing of multiple input channels is still supported by the experiments. This adds to the existing state-of-the-art in photonic tensor processors. In my opinion, the manuscript is suitable to be published.

We'd like to thank, deeply in heart, all reviewers who paid substantial efforts on this manuscript. Their professional and insightful comments really help us improve the quality of the manuscript significantly. The responses to all concerns are described one by one as follows:

To Reviewer #1:

Black: Comments from the reviewer.

Blue: Response of the authors.

Red underlined: Revisions to the manuscript.

I have reviewed a previous version of this manuscript (Reviewer 1). My recommendation was to request further modifications in the manuscript in the intent of balancing claims and evidence. The authors made significant changes to the manuscript and addressed most of my concerns satisfactorily. I would like to thank the authors for the detailed response letter, once again.

As I expected, though the addressed comments improved the quality of the manuscript, it reduced the overall impact of the paper, because the 1. proof-of-concept experiment does not outperform electronics (though he proposed architecture might), 2. the proposed speed was technically achieved (20GBaud) but since the operation was not done in real-time, this number is misleading relative to digital processors; and 3. the overhead introduced by the encoding/decoding scheme can become prohibitive to a practical implementation that outcompetes electronics.

With that said, thanks to the modifications in the manuscript, the careful reader will adequately understand the limitations of the PTFP. If the editor is comfortable with this changing scope, I will recommend publication after addressing minor issues below:

Response:

We would like thank the reviewer for the helpful comments again. The issues are addressed in the following sections.

#1

In your calculation for the ADC/DAC power consumption (Response #5), you used a FOM formula (from Murmann 2015) to estimate how power scales based on a demonstrated device. I'd like to point out that the formula is not directly applicable to speeds above 100MHz (see Fig. 3 in Murmann 2015). Power scales superlinearly with frequency beyond that point. So your

calculations are off.

Response:

The reviewer is appreciated for pointing out the problem of our previous ADC energy calculation, so that we can strengthen the evidence of our manuscript. The reviewer is right. The Fig. 3 in Murmann's article indeed describes that it is more difficult to achieve high FoM at high speed than that at low speed.

To revise our calculation in a rigorous way, we have observed more typical published high-speed ADCs for reference, including a 5 GS/s ADC consuming 29 mW power (M. Guo, et al, IEEE Access, 2020, doi: 10.1109/ACCESS.2020.3012699), a 10 GS/s ADC consuming 50.8 mW power (M. Zhang, et al, IEEE JSSC, 2020, doi: 10.1109/JSSC.2020.3012776), and a 32 GS/s ADC consuming 199 mW power (L. Kull, et al, IEEE VLSI symposium, 2018, doi: 10.1109/VLSIC.2018.8502268). The power consumption grows approximately linearly, but the SNDR drops from 48.5 dB to around 43 dB. Therefore, at high speed, the expected power consumption may not vary significantly but the signal-to-noise ratio will be lower for several decibels.

To address this issue in our manuscript, revisions are done as follows.

(Lines 14-25, Page 7 in Suppl.): “For ADCs, several high-speed ADCs [S7-S9] are referenced, including a 5 GS/s ADC consuming 29 mW, a 10 GS/s ADC consuming 50.8 mW, and a 32 GS/s ADC consuming 199 mW. The power consumption grows approximately linearly with SNDR drops from 48.5 dB to around 43 dB. It infers that a 20 GS/s ADC may consume the power of ~120 mW at 7-bit effective resolution. This estimation is also consistent with the observation in a review of ADCs [S10]. Then, the A/D interconversion of a symbol introduces $(120+5.53)/20=6.28$ pJ overhead. ...Assume a $[3\times 3, 32, 32]$ kernel is implemented on-chip (technically feasible), the energy overhead for a single operation can be calculated as

$$E = \left(\frac{(120 + 5.53) \text{ [mW]}}{20 \text{ Gbaud} \times 32 \times 9} + \frac{7960 \text{ [mW]}}{20 \text{ Gbaud} \times 32 \times 32 \times 9} + \frac{0.317 \text{ [pJ]}}{32 \times 9} \right) / 2 = 0.0329 \text{ pJ. } ”$$

S7. M. Guo, J. Mao, S. Sin, H. Wei, and R. Martins, A 5 GS/s 29 mW interleaved SAR ADC with 48.5 dB SNDR using digital-mixing background timing-skew calibration for direct sampling applications, IEEE Access 8, 138954 (2020).

S8. M. Zhang, Y. Zhu, C. Chan, and R. Martins, An 8-Bit 10-GS/s $16\times$ interpolation-based time-domain ADC with <1.5 -ps uncalibrated quantization steps, IEEE Journal of Solid-State Circuits 55, 3225-3235 (2020).

S9. L. Kull, D. Luu, C. Menolfi, T. Morf, P. Francese, M. Braendli, M. Kossel, A. Cevrero, I. Ozkaya, and T. Toifl, A 10-bit 20-40 GS/s ADC with 37 dB SNDR at 40 GHz input using first order sampling bandwidth calibration, IEEE Symposium on VLSI Circuits (2018).

(Lines 1-4, Page 9): “we show that the high-speed DACs and ADCs will introduce an energy

overhead of around 6.28 pJ/symbol. With the expansion of the integration scale, it is promising to dilute the energy overhead of DACs and ADCs down to 0.011 pJ per operation.”

#2

I'd like more justification on why the laser power can be scaled with C_{in} . It looks like in your analysis you fix total laser power to 20dBm, so if C_{in} is scaled up to 32 taps, will that provide sufficient optical power to reach appropriate SNR at the photodetector?

Response:

The reviewer's question is highly appreciated. The answer to the question is Yes. The scaled optical power is able to achieve sufficient SNR at photodetection.

As the total power of input light is limited by the nonlinear effect of Si_3N_4 waveguide, the optical power of every laser is scaled with C_{in} . At photodetection, the photoelectric current is generated by the total optical power of all wavelengths. Therefore, as long as the total optical power is fixed at 20 dBm, the SNR is not influenced by the scaling of C_{in} .

To address this issue in the manuscript, revisions are done as follows.

(Lines 21-22, Page 7 in Suppl.): “We note that the SNR of photodetection is not influenced by the number of input channels with fixed input optical powr. Ref. [33] presents the details.”

To Reviewer #3:

Black: Comments from the reviewer.

Blue: Response of the authors.

Red underlined: Revisions to the manuscript.

I have reviewed previous versions of this manuscript, pointing out several aspects that need to be revised. The major issues were in the claims made by the authors and the experimental evidence provided, related to the size of the kernel during the PTFP operation. The main novelty was claimed to be the ability of the proposed PTFP to perform higher dimensional tensor operation in hardware utilizing the properties of the photonics chip (wavelength, space, time multiplexing). However, the experiment was performed with a reduced set of dimensions. Furthermore, various other critical points were raised in previous review reports.

The authors have thoroughly addressed the raised issues. The current manuscript has adapted the claims and added explicit explanations on the reduced dimensionality of the PTFP operation in experiment. The authors clarify that the experiment acts as a proof-of-concept and reduce their claim to the proposition of such a PTFP concept. The authors have added additional information on power consumption, delay line scaling, data processing among others which raises the quality of the current manuscript.

With the reduced claim, the impact of the manuscript is also reduced. However, the main novelty to exploit different wavelengths for the processing of multiple input channels is still supported by the experiments. This adds to the existing state-of-the-art in photonic tensor processors. In my opinion, the manuscript is suitable to be published.

Response:

We greatly appreciate the reviewer's efforts paid on our manuscript for several rounds. We are so happy to see the recommendation of publication.